# Do SSL Models Have Déjà Vu? A Case of Unintended Memorization in Self-supervised Learning

**Casey Meehan**[*,1], **Florian Bordes**[*,2,3], **Pascal Vincent**[2,3]
, **Kamalika Chaudhuri**[†,1,3], **Chuan Guo**[†,3]
[1]UCSD, [2]Mila, Université de Montréal, [3]FAIR, Meta
[*]Equal contribution, [†]Equal direction contribution
cmeehan@eng.ucsd.edu, florian.bordes@umontreal.ca,
{pascal,kamalika,chuanguo}@meta.com

## Abstract

Self-supervised learning (SSL) algorithms can produce useful image representations by learning to associate different parts of natural images with one another. However, when taken to the extreme, SSL models can unintendedly memorize *specific* parts in individual training samples rather than learning semantically meaningful associations. In this work, we perform a systematic study of the unintended memorization of image-specific information in SSL models—which we refer to as *déjà vu memorization*. Concretely, we show that given the trained model and a crop of a training image containing only the background (*e.g.*, water, sky, grass), it is possible to infer the foreground object with high accuracy or even visually reconstruct it. Furthermore, we show that *déjà vu* memorization is common to different SSL algorithms, is exacerbated by certain hyperparameter choices, and cannot be detected by conventional techniques for evaluating representation quality. Our study of *déjà vu* memorization reveals previously unknown privacy risks in SSL models, as well as suggests potential practical mitigation strategies.

## 1 Introduction

Self-supervised learning (SSL) [11; 12; 33; 2; 9; 20] aims to learn general representations of content-rich data without explicit labels by solving a *pretext task*. In many recent works, such pretext tasks rely on joint-embedding architectures whereby randomized image augmentations are applied to create multiple views of a training sample, and the model is trained to produce similar representations for those views. When using cropping as random image augmentation, the model learns to associate objects or parts (including the background scenery) that co-occur in an image. However, doing so also arguably exposes the training data to higher privacy risk as objects in training images can be explicitly memorized by the SSL model. For example, if the training data contains the photos of individuals, the SSL model may learn to associate the face of a person with their activity or physical location in the photo. This may allow an adversary to extract such information from the trained model for targeted individuals.

In this work, we aim to evaluate to what extent SSL models memorize the association of specific objects in training images or the association of objects and their specific backgrounds, and whether this memorization signal can be used to reconstruct the model's training samples. Our results demonstrate that SSL models memorize such associations beyond simple correlation. For instance, in Figure 1 (**left**), we use the SSL representation of a *training image crop containing only water* and this enables us to reconstruct the object in the foreground with remarkable specificity—in this case a black swan. By contrast, in Figure 1 (**right**), when using the *crop from the background of a test set image* that the SSL model *has not seen before*, its representation only contains enough information to

37th Conference on Neural Information Processing Systems (NeurIPS 2023).

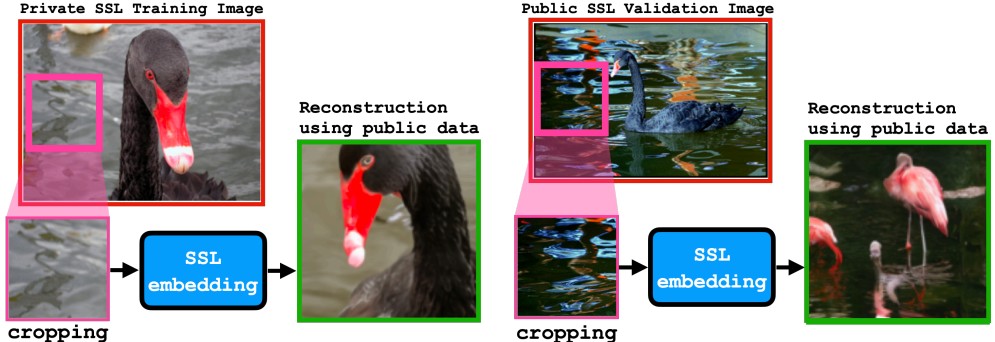

**Figure 1: Left:** Reconstruction of an SSL training image from a crop containing only the background. The SSL model memorizes the association of this *specific* patch of water (pink square) to this *specific* foreground object (a black swan) in its embedding, which we decode to visualize the full training image. **Right:** The reconstruction technique fails on a public test image that the SSL model has not seen before.

infer, through correlation, that the foreground object was likely some kind of waterbird — but not the specific one in the image.

Figure 1 shows that SSL models suffer from the unintended memorization of images in their training data—a phenomenon we refer to as *déjà vu memorization* [1] Beyond visualizing *déjà vu* memorization through data reconstruction, we also design a series of experiments to quantify the degree of memorization for different SSL algorithms, model architectures, training set size, *etc*. We observe that *déjà vu* memorization is exacerbated by the atypically large number of training epochs often recommended in SSL training, as well as certain hyperparameters in the SSL training objective. Perhaps surprisingly, we show that *déjà vu* memorization occurs even when the training set is large—as large as half of ImageNet [13]—and can continually worsen even when standard techniques for evaluating learned representation quality (such as linear probing) do not suggest increased overfitting. Our work serves as the first systematic study of unintended memorization in SSL models and motivates future work on understanding and preventing this behavior. Specifically, we:

- Elucidate how SSL representations memorize aspects of individual training images, what we call *déjà vu* memorization;

- Design a novel training data reconstruction pipeline for non-generative vision models. This is in contrast to many prominent reconstruction algorithms like [7; 8], which rely on the model itself to generate its own memorized samples and is not possible for SSL models or classifiers;

- Propose metrics to quantify the degree of *déjà vu* memorization committed by an SSL model. This allows us to observe how *déjà vu* changes with training epochs, dataset size, training criteria, model architecture and more.

## 2  Preliminaries and Related Work

**Self-supervised learning** (SSL) is a machine learning paradigm that leverages unlabeled data to learn representations. Many SSL algorithms rely on *joint-embedding* architectures (*e.g.*, SimCLR [11], Barlow Twins [33], VICReg [2] and Dino [10]), which are trained to associate different augmented views of a given image. For example, in SimCLR, given a set of images $\mathcal{A} = \{A_1, \ldots, A_n\}$ and a randomized augmentation function $\mathrm{aug}$, the model is trained to maximize the cosine similarity of draws of $\mathrm{SSL}(\mathrm{aug}(A_i))$ with each other and minimize their similarity with $\mathrm{SSL}(\mathrm{aug}(A_j))$ for $i \neq j$. The augmentation function $\mathrm{aug}$ typically consists of operations such as cropping, horizontal flipping, and color transformations to create different views that preserve an image's semantic properties.

**SSL representations.**  Once an SSL model is trained, its learned representation can be transferred to different downstream tasks. This is often done by extracting the representation of an image from

---

[1]The French loanword *déjà vu* means 'already-seen', just as an image is seen and memorized in training.

the *backbone model*[2] and either training a linear probe on top of this representation or finetuning the backbone model with a task-specific head [3]. It has been shown that SSL representations encode richer visual details about input images than supervised models do [4]. However, from a privacy perspective, this may be a cause for concern as the model also has more potential to overfit and memorize precise details about the training data compared to supervised learning. We show concretely that this privacy risk can indeed be realized by defining and measuring *déjà vu* memorization.

**Privacy risks in ML.**   When a model is overfit on privacy-sensitive data, it memorizes specific information about its training examples, allowing an adversary with access to the model to learn private information [30; 16]. Privacy attacks in ML range from the simplest and best-studied *membership inference attacks* [26; 25; 24] to *attribute inference* [17; 22; 21] and *data reconstruction* [7; 1; 19] attacks. In the former, the adversary only infers whether an individual participated in the training set. Our study of *déjà vu* memorization is most similar to the latter: we leverage SSL representations of the training image background to infer and reconstruct the foreground object. In another line of work in the NLP domain [6; 7]: when prompted with a context string present in the training data, a large language model is shown to generate the remainder of string at test time, revealing sensitive text like home addresses. This method was recently extended to extract memorized images from Stable Diffusion [8]. We exploit memorization in a similar manner: given partial information about a training sample, the model is prompted to reveal the rest of the sample.[3] In our case, however, since the SSL model is not generative, extraction is significantly harder and requires careful design.

## 3   Defining *Déjà Vu* Memorization

**What is *déjà vu* memorization?**   At a high level, the objective of SSL is to learn general representations of objects that occur in nature. This is often accomplished by associating different parts of an image with one another in the learned embedding. Returning to our example in Figure 1, given an image whose background contains a patch of water, the model may learn that the foreground object is a water animal such as duck, pelican, otter, *etc.*, by observing different images that contain water from the training set. We refer to this type of learning as *correlation*: the association of objects that tend to co-occur in images from the training data distribution.

A natural question to ask is *"Can the reconstruction of the black swan in Figure 1 be reasoned as correlation?"* The intuitive answer may be no, since the reconstructed image is qualitatively very similar to the original image. However, this reasoning implicitly assumes that for a random image from the training data distribution containing a patch of water, the foreground object is unlikely to be a black swan. Mathematically, if we denote by $\mathcal{P}$ the training data distribution and $A$ the image, then

$$p_{\text{corr}} := \mathbb{P}_{A \sim \mathcal{P}}(\text{object}(A) = \texttt{black swan} \mid \text{crop}(A) = \texttt{water})$$

is the probability of inferring that the foreground object is a black swan through *correlation*. This probability may be naturally high due to biases in the distribution $\mathcal{P}$, *e.g.*, if $\mathcal{P}$ contains no other water animal except for black swans. In fact, such correlations are often exploited to learn a model for image inpainting with great success [32; 27].

Despite this, we argue that reconstruction of the black swan in Figure 1 is *not* due to correlation, but rather due to *unintended memorization*: the association of objects unique to a single training image. As we will show in the following sections, the example in Figure 1 is not a rare success case and can be replicated across many training samples. More importantly, failure to reconstruct the foreground object in Figure 1 (**right**) on test images hints at inferring through correlation is unlikely to succeed—a fact that we verify quantitatively in Section 4.1. Motivated by this discussion, we give a verbal definition of *déjà vu* memorization below, and design a testing methodology to quantify *déjà vu* memorization in Section 3.1.

---

[2]SSL methods often use a trick called *guillotine regularization* [3], which decomposes the model into two parts: a *backbone model* and a *projector* consisting of a few fully-connected layers. Such trick is needed to handle the misalignment between the pretext SSL task and the downstream task.

[3]We recognize that it is easier to find a context string that might have been in the training data of a large language models that finding the exact pixels that constitutes a crop of a training image. However, this paper focus on revealing a memorization phenomena in SSL and does not aim to provide a complete picture of all the privacy risk that it might entails.

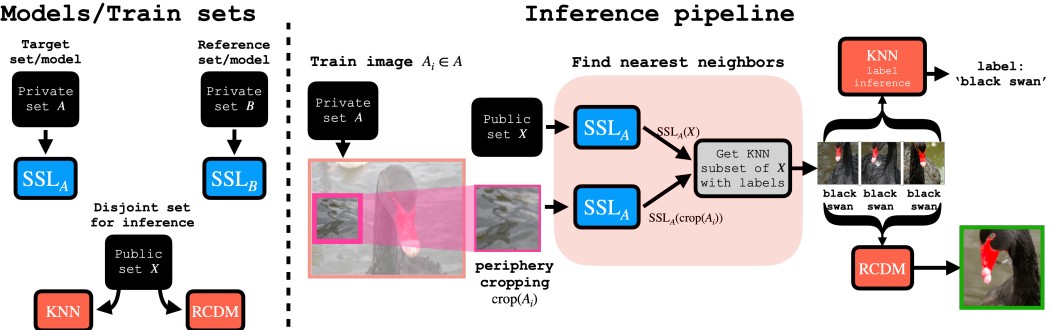

**Figure 2:** Overview of testing methodology. **Left:** Data is split into *target set* $\mathcal{A}$, *reference set* $\mathcal{B}$ and *public set* $\mathcal{X}$ that are pairwise disjoint. $\mathcal{A}$ and $\mathcal{B}$ are used to train two SSL models $\mathrm{SSL}_A$ and $\mathrm{SSL}_B$ in the same manner. $\mathcal{X}$ is used for KNN decoding at test time. **Right:** Given a training image $A_i \in \mathcal{A}$, we use $\mathrm{SSL}_A$ to embed $\mathrm{crop}(A_i)$ containing only the background, as well as the entire set $\mathcal{X}$ and find the $k$-nearest neighbors of $\mathrm{crop}(A_i)$ in $\mathcal{X}$ in the embedding space. These KNN samples can be used directly to infer the foreground object (*i.e.*, class label) in $A_i$ using a KNN classifier, to be visualized directly or their embeddings can be averaged as input to the trained RCDM to visually reconstruct the image $A_i$. For instance, the KNN visualization results in Figure 1 (left) when given $\mathrm{SSL}_A(\mathrm{crop}(A_i))$ and results in Figure 1 (right) when given $\mathrm{SSL}_A(\mathrm{crop}(B_i))$ for an image $B_i \in \mathcal{B}$.

> **Definition:** A model exhibits *déjà vu memorization* when it retains information so specific to an individual training image, that it enables recovery of aspects particular to that image given a part that does not contain them. The recovered aspect must be beyond what can be inferred using only correlations in the data distribution.

We intentionally kept the above definition broad enough to encompass different types of information that can be inferred about the training image, including but not restricted to object category, shape, color and position. For example, if one can infer that the foreground object is red given the background patch with accuracy significantly beyond correlation, we consider this an instance of *déjà vu* memorization as well. We mainly focus on object category to quantify *déjà vu* memorization in Section 4 since the ground truth label can be easily obtained. We consider other types of information more qualitatively in the visual reconstruction experiments in Section 5.

**Distinguishing memorization from correlation.** When measuring *déjà vu* memorization, it is crucial to differentiate what the model associates through *memorization* and what it associates through *correlation*. Our testing methodology is based on the following intuitive definition.

> **Definition:** If an SSL model associates two parts in a training image, we say that it is due to *correlation* if other SSL models trained on a similar dataset from $\mathcal{P}$ without this image would likely make the same association. Otherwise, we say that it is due to *memorization*.

Notably, such intuition forms the basis for differential privacy (DP; Dwork et al. [15]; Dwork & Roth [14])—the most widely accepted notion of privacy in ML.

### 3.1 Testing Methodology for Measuring *Déjà Vu* Memorization

In this section, we use the above intuition to measure the extent of *déjà vu* memorization in SSL. Figure 2 gives an overview of our testing methodology.

**Dataset splitting.** We focus on testing *déjà vu* memorization for SSL models trained on the ImageNet dataset [13][4]. Our test first splits the ImageNet training set into three independent and disjoint subsets $\mathcal{A}$, $\mathcal{B}$ and $\mathcal{X}$. The dataset $\mathcal{A}$ is called the *target set* and $\mathcal{B}$ is called the *reference set*. The two datasets are used to train two separate SSL models, $\mathrm{SSL}_A$ and $\mathrm{SSL}_B$, called the *target model* and the *reference model*. Finally, the dataset set $\mathcal{X}$ is used as an auxiliary public dataset to extract information from $\mathrm{SSL}_A$ and $\mathrm{SSL}_B$. Our dataset splitting serves the purpose of

---

[4]We used the face-blurred version of ImageNet for privacy purposes.

distinguishing memorization from correlation in the following manner. Given a sample $A_i \in \mathcal{A}$, if our test returns the same result on $\text{SSL}_A$ and $\text{SSL}_B$ then it is likely due to correlation because $A_i$ is not a training sample for $\text{SSL}_B$. Otherwise, because $\mathcal{A}$ and $\mathcal{B}$ are drawn from the same underlying distribution, our test must have inferred some information unique to $A_i$ due to memorization. Thus, by comparing the difference in the test results for $\text{SSL}_A$ and $\text{SSL}_B$, we can measure the degree of *déjà vu* memorization[5].

**Extracting foreground and background crops.**    Our testing methodology aims at measuring what can be inferred about the foreground object in an ImageNet sample given a background crop. This is made possible because ImageNet provides bounding box annotations for a subset of its training images—around 150K out of 1.3M samples. We split these annotated images equally between $\mathcal{A}$ and $\mathcal{B}$. Given an annotated image $A_i$, we treat everything inside the bounding box as the foreground object associated with the image label, denoted $\text{object}(A_i)$. We take the largest possible crop that does not intersect with any bounding box as the background crop (or *periphery crop*), denoted $\text{crop}(A_i)$[6]

**KNN-based test design.**    Joint-embedding SSL approaches encourage the embeddings of random crops of a training image $A_i \in \mathcal{A}$ to be similar. Intuitively, if the model exhibits *déjà vu* memorization, it is reasonable to expect that the embedding of $\text{crop}(A_i)$ is similar to that of $\text{object}(A_i)$ since both crops are from the same training image. In other words, $\text{SSL}_A(\text{crop}(A_i))$ encodes information about $\text{object}(A_i)$ that cannot be inferred through correlation. However, decoding such information is challenging as these approaches do not learn a decoder associated with the encoder $\text{SSL}_A$.

Here, we leverage the public set $\mathcal{X}$ to decode the information contained in $\text{crop}(A_i)$ about $\text{object}(A_i)$. More specifically, we map images in $\mathcal{X}$ to their embeddings using $\text{SSL}_A$ and extract the $k$-nearest-neighbor (KNN) subset of $\text{SSL}_A(\text{crop}(A_i))$ in $\mathcal{X}$. We can then decode the information contained in $\text{crop}(A_i)$ in one of two ways:

- *Label inference:* Since $\mathcal{X}$ is a subset of ImageNet, each embedding in the KNN subset is associated with a class label. If $\text{crop}(A_i)$ encodes information about the foreground object, its embedding will be close to samples in $\mathcal{X}$ that have the same class label (*i.e.*, foreground object category). We can then use a KNN classifier to infer the foreground object in $A_i$ given $\text{crop}(A_i)$.
- *Visualization:* Since we have access to a KNN subset associated to a given $\text{crop}(A_i)$, we can visualize directly the images associated to this subset. Then, we can infer through visualizing what is common within this subset, what information can be retrieved for this single crop. In addition, to simplify the visualization pipeline and to map directly a given crop representation to an image, we train an RCDM [4]—a conditional generative model—on $\mathcal{X}$ to decode $\text{SSL}_A$ embeddings into images. The RCDM reconstruction can recover qualitative aspects of an image remarkably well, such as recovering object color or spatial orientation using its SSL embedding. Given the KNN subset, we average their SSL embeddings and use the RCDM model to visually reconstruct $A_i$.

In Section 4, we focus on quantitatively measuring *déjà vu* memorization with label inference, and then use the KNN to visualize *déjà vu* memorization in Section 5.

## 4    Quantifying *Déjà Vu* Memorization

We apply our testing methodology to quantify a specific form of *déjà vu* memorization: inferring the foreground object (class label) given a crop of the background.

**Extracting model embeddings.**    We test *déjà vu* memorization on a variety of popular SSL algorithms, with a focus on VICReg [2]. These algorithms produce two embeddings given an input image: a *backbone* embedding and a *projector* embedding that is derived by applying a small fully-connected network on top of the backbone embedding. Unless otherwise noted, all SSL embeddings refer to the projector embedding. To understand whether *déjà vu* memorization is particular to SSL, we also evaluate embeddings produced by a supervised model $\text{CLF}_A$ trained on $\mathcal{A}$. We apply the same set of image augmentations as those used in SSL and train $\text{CLF}_A$ using the cross-entropy loss to predict ground truth labels.

---

[5]See Appendix A.2.1 for details on how the dataset splits are generated.

[6]We also present another heuristic in Appendix A.8 which takes a corner crop as the background crop, allowing our test to be run without bounding box annotations.

**Identifying the most memorized samples.** Prior works have shown that certain training samples can be identified as more prone to memorization than others [16; 28; 29]. Similarly, we provide a heuristic to identify the most memorized samples in our label inference test using confidence of the KNN prediction. Given a periphery crop, $\text{crop}(A_i)$, let $\text{KNN}_A\big(\text{crop}(A_i)\big) \subseteq \mathcal{X}$ denote its $k$-nearest neighbors in the embedding space of $\text{SSL}_A$. From this KNN subset we can obtain: **(1)** $\text{KNN}_A^{\text{prob}}\big(\text{crop}(A_i)\big)$, the vector of class probabilities (normalized counts) induced by the KNN subset, and **(2)** $\text{KNN}_A^{\text{conf}}\big(\text{crop}(A_i)\big)$, the negative entropy of the probability vector $\text{KNN}_A^{\text{prob}}\big(\text{crop}(A_i)\big)$, as confidence of the KNN prediction. When entropy is low, the neighbors agree on the class of $A_i$ and hence confidence is high. We can sort the confidence score $\text{KNN}_A^{\text{conf}}\big(\text{crop}(A_i)\big)$ across samples $A_i$ in decreasing order to identify the most confidently predicted samples, which likely correspond to the most memorized samples when $A_i \in \mathcal{A}$.

## 4.1 Population-level Memorization

Our first measure of *déjà vu* memorization is population-level label inference accuracy: *What is the average label inference accuracy over a subset of SSL training images given their periphery crops?* To understand how much of this accuracy is due to $\text{SSL}_A$'s *déjà vu* memorization, we compare with a correlation baseline using the reference model: $\text{KNN}_B$'s label inference accuracy on images $A_i \in \mathcal{A}$. In principle, this inference accuracy should be significantly above chance level ($1/1000$ for ImageNet) because the periphery crop may be highly indicative of the foreground object through correlation, *e.g.*, if the periphery crop is a basketball player then the foreground object is likely a basketball. Figure 3 compares the accuracy of $\text{KNN}_A$ to that of $\text{KNN}_B$ when inferring the labels of images in $A_i \in \mathcal{A}^7$ using $\text{crop}(A_i)$. Results are shown for VICReg and the supervised model; trends for other models are shown in Appendix A.5. For both VICReg and supervised models, inferring the class of $\text{crop}(A_i)$ using $\text{KNN}_B$ (dashed line) through correlation achieves a reasonable accuracy that is significantly above chance level.

However, for VICReg, the inference accuracy using $\text{KNN}_A$ (solid red line) is significantly higher, and the accuracy gap between $\text{KNN}_A$ and $\text{KNN}_B$ indicates the degree of *déjà vu* memorization. We highlight two observations:

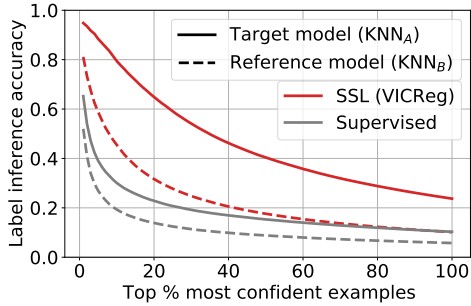

- The accuracy gap of VICReg is significantly larger than that of the supervised model. This is especially notable when accounting for the fact that the supervised model is trained to associate randomly augmented crops of images with their ground truth labels. In contrast, VICReg has no label access during training but the embedding of a periphery crop can still encode the image label.
- For VICReg, inference accuracy on the $1\%$ most confident examples is nearly $95\%$, which shows that our simple confidence heuristic can effectively identify the most memorized samples. This result suggests that an adversary can use this heuristic to identify vulnerable training samples to launch a more focused privacy attack.

**Figure 3:** Accuracy of label inference using the target model (trained on $\mathcal{A}$) vs. the reference model (trained on $\mathcal{B}$) on the top $\%$ most confident examples $A_i \in \mathcal{A}$ using only $\text{crop}(A_i)$. For VICReg, there is a large accuracy gap between the two models, indicating a significant degree of *déjà vu* memorization.

**The *déjà vu* score.** The curves of Figure 3 show memorization across confidence values for a single training scenario. To study how memorization changes with different hyperparamters, we extract a single value from these curves: the *déjà vu score* at confidence level $p$. In Figure 3, this is the gap between the solid red (or gray) and dashed red (or gray) where confidence ($x$-axis) equal $p\%$. In other words, given the periphery crops of set $\mathcal{A}$, $\text{KNN}_A$ and $\text{KNN}_B$ separately select and label their top $p\%$ most confident examples, and we report the difference in their accuracy. The *déjà vu* score captures both the degree of memorization by the accuracy gap and the *ability to identify memorized examples* by the confidence level. If the score is 10% for $p = 33\%$, $\text{KNN}_A$ has 10%

---

[7]The sets $\mathcal{A}$ and $\mathcal{B}$ are exchangeable, and in practice we repeat this test on images from $\mathcal{B}$ using $\text{SSL}_B$ as the target model and $\text{SSL}_A$ as the reference model, and average the two sets of results.

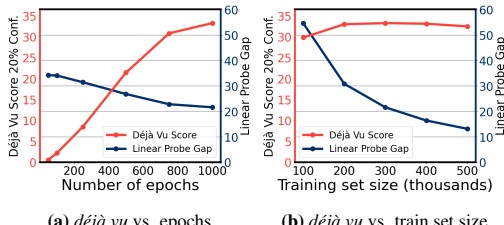

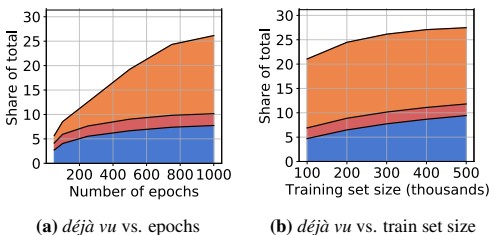

**Figure 4:** Effect of training epochs and train set size on *déjà vu* score (red) in comparison with linear probe accuracy train-test gap (dark blue) for VICReg. **Left:** *déjà vu* score increases with training epochs, indicating growing memorization while the linear probe baseline decreases significantly. **Right:** *déjà vu* score stays roughly constant with training set size while linear probe gap shrinks significantly, suggesting that memorization may be problematic even for large datasets.

**Figure 5:** Partition of samples $A_i \in \mathcal{A}$ into the four categories: unassociated (not shown), memorized, misrepresented and correlated for VICReg. The memorized samples—those whose labels are predicted by $\text{KNN}_A$ but not by $\text{KNN}_B$—occupy a significantly larger share of the training set than the misrepresented samples—those predicted by $\text{KNN}_B$ but not $\text{KNN}_A$ by chance.

higher accuracy on its most confident third of $\mathcal{A}$ than $\text{KNN}_B$ does on its most confident third. In the following, we set $p = 20\%$, approximately the largest gap for VICReg (red lines) in Figure 3.

**Comparison with the linear probe train-test gap.** A standard method for measuring SSL performance is to train a linear classifier—what we call a 'linear probe'—on its embeddings and compute its performance on a held out test set. From a learning theory standpoint, one might expect the linear probe's train-test accuracy gap to be indicative of memorization: the more a model overfits, the larger is the difference between train set and test set accuracy. However, as seen in Figure 4, the linear probe gap (dark blue) fails to reveal memorization captured by the *déjà vu* score (red) [8].

## 4.2 Sample-level Memorization

The *déjà vu* score shows, *on average*, how much better an adversary can select and classify images when using the target model trained on them. This average score does not tell us how many individual images have their label successfully recovered by $\text{KNN}_A$ but not by $\text{KNN}_B$. In other words, how many images are exposed by virtue of *being in training set* $\mathcal{A}$: a risk notion foundational to differential privacy. To better quantify what fraction of the dataset is at risk, we perform a sample-level analysis by fixing a sample $A_i \in \mathcal{A}$ and observing the label inference result of $\text{KNN}_A$ vs. $\text{KNN}_B$. To this end, we partition samples $A_i \in \mathcal{A}$ based on the result of label inference into four distinct categories: **Unassociated** - label inferred with neither KNN; **Memorized** - label inferred only with $\text{KNN}_A$; **Misrepresented** - label inferred only with $\text{KNN}_B$; **Correlated** - label inferred with both KNNs.

Intuitively, unassociated samples are ones where the embedding of $\text{crop}(A_i)$ does not encode information about the label. Correlated samples are ones where the label can be inferred from $\text{crop}(A_i)$ using correlation, *e.g.*, inferring the foreground object is basketball given a crop showing a basketball player. Ideally, the misrepresented set should be empty but contains a small portion of examples due to chance. *Déjà vu* memorization occurs for memorized samples where the embedding of $\text{SSL}_B$ does not encode the label but the embedding of $\text{SSL}_A$ does. To measure the pervasiveness of *déjà vu* memorization, we compare the size of the memorized and misrepresented sets. Figure 5 shows how the four categories of examples change with number of training epochs and training set size. The unassociated set is not shown since the total share adds up to one. The misrepresented set remains under $5\%$ and roughly unchanged across all settings, consistent with our explanation that it is due to chance. In comparison, VICReg's memorized set surpasses $15\%$ at 1000 epochs. Considering that up to $5\%$ of these memorized examples could also be due to chance, we conclude that **at least 10% of VICReg's training set is *déjà vu* memorized.**

---

[8]See section 6 for further discussion of the *déjà vu* score trends of Figure 4.

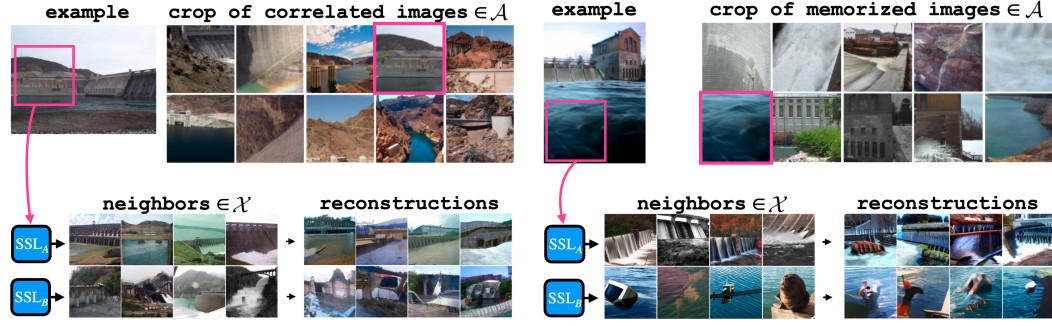

**(a)** A correlated dam example

**(b)** A memorized dam example

**Figure 6:** Correlated and Memorized examples from the *dam* class. Both $SSL_A$ and $SSL_B$ are SimCLR models. **Left:** The periphery crop (pink square) contains a concrete structure that is often present in images of dams. Consequently, the KNN can infer the foreground object using representations from both $SSL_A$ and $SSL_B$ through this correlation. **Right:** The periphery crop only contains a patch of water. The embedding produced by $SSL_B$ only contains enough information to infer that the foreground object is related to water, as reflected by its KNN set. In contrast, the embedding produced by $SSL_A$ memorizes the association of this patch of water with dam and the KNN select images of dams.

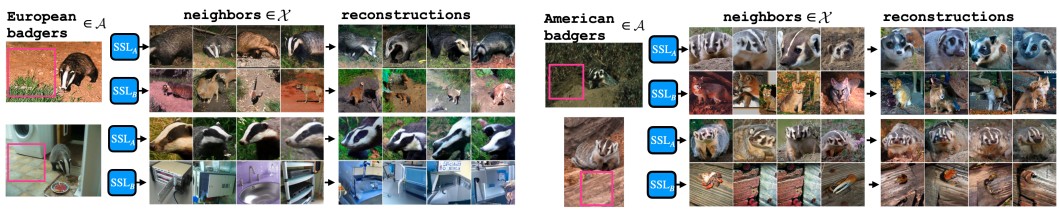

**(a)** Memorized European badgers

**(b)** Memorized American badgers

**Figure 7:** Visualization of *déjà vu* memorization beyond class label. Both $SSL_A$ and $SSL_B$ are VICReg models. The four images shown belong to the memorized set of $SSL_A$ from the *badger* class. KNN images using embeddings from $SSL_A$ can reveal not only the correct class label, but also the specific badger species: *European* (left) and *American* (right). Such information does not appear to be memorized by the reference model $SSL_B$.

# 5 Visualizing *Déjà Vu* Memorization

Beyond enabling label inference using a periphery crop, we show that *déjà vu* memorization allows the SSL model to encode other forms of information about a training image. Namely, we leverage an external public dataset $\mathcal{X}$ and use it to find the nearest neighborhoods in this public dataset given a training periphery crop. We aim to answer the following two questions: **(1)** Can we visualize the distinction between correlation and *déjà vu* memorization? **(2)** What foreground object details can be extracted from the SSL model beyond class label?

**Public image retrieval pipeline**    Following the pipeline in Figure 2, we use the projector embedding to find the KNN subset for the periphery crop, $\mathrm{crop}(A_i)$, and visualize the images belonging to this KNN subset.

**RCDM pipeline.**    RCDM is a conditional generative model that is trained on the *backbone embedding* of images $X_i \in \mathcal{X}$ to generate an image that resembles $X_i$. At test time, following the pipeline in Figure 2, we first use the projector embedding to find the KNN subset for the periphery crop, $\mathrm{crop}(A_i)$, and then average their backbone embeddings as input to the RCDM model. Then, RCDM decodes this representation to visualize its content.

**Visualizing Correlation vs. Memorization.**    Figure 6 shows examples of dams from the correlated set (left) and the memorized set (right) as defined in Section 4.2, along with the associated KNN set. In Figure 6a, the periphery crop is represented by the pink square, which contains concrete structure attached to the dam's main structure. As a result, both $SSL_A$ and $SSL_B$ produce embeddings of

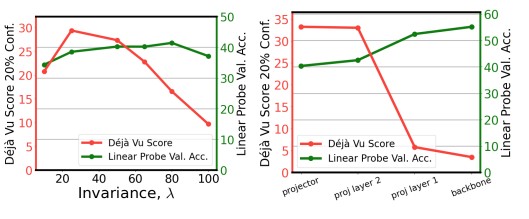

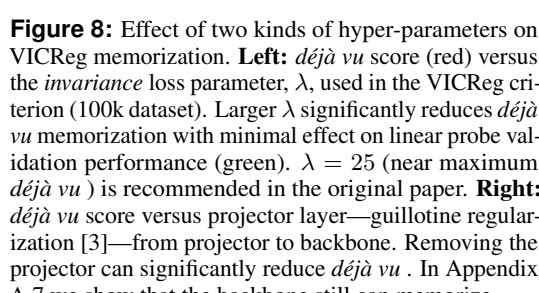

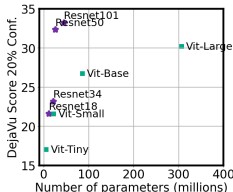

**(a)** Loss hyper-parameter     **(b)** Guillotine regularization     **(a)** *déjà vu* vs. capacity

| Criteria | DV | Acc P/B |
|---|---|---|
| Supervised | 8.9 | 55.3/61.1 |
| Byol[18] | 8.0 | 54.3/59.4 |
| SimCLR[11] | 10.0 | 44.2/54.1 |
| Dino[10] | 14.5 | 26.3/55.7 |
| Barlow T.[33] | 30.5 | 33.7/54.4 |
| VICReg[2] | **33.2** | 40.3/55.2 |

**(b)** *déjà vu* (DV) vs. Criterion

**Figure 8:** Effect of two kinds of hyper-parameters on VICReg memorization. **Left:** *déjà vu* score (red) versus the *invariance* loss parameter, $\lambda$, used in the VICReg criterion (100k dataset). Larger $\lambda$ significantly reduces *déjà vu* memorization with minimal effect on linear probe validation performance (green). $\lambda = 25$ (near maximum *déjà vu*) is recommended in the original paper. **Right:** *déjà vu* score versus projector layer—guillotine regularization [3]—from projector to backbone. Removing the projector can significantly reduce *déjà vu*. In Appendix A.7 we show that the backbone still can memorize.

**Figure 9:** Effect of model architecture and criterion on *déjà vu* memorization. **Left:** *déjà vu* score with VICReg for resnet (purple) and vision transformer (green) architectures versus number of model parameters. As expected, memorization grows with larger model capacity. This trend is more pronounced for convolutional (resnet) than transformer (ViT) architectures. **Right:** Comparison of *déjà vu* score 20% conf. and ImageNet linear probe validation accuracy (P: using projector embeddings, B: using backbone embeddings) for various SSL criteria.

$\text{crop}(A_i)$ whose KNN set in $\mathcal{X}$ consist of dams, *i.e.*, there is a correlation between the concrete structure in $\text{crop}(A_i)$ and the foreground dam. In Figure 6b, the periphery crop only contains a patch of water, which does not strongly correlate with dams in the ImageNet distribution. Evidently, the reference model $\text{SSL}_B$ embeds $\text{crop}(A_i)$ close to that of other objects commonly found in water, such as sea turtle and submarine. In contrast, the KNN set according to $\text{SSL}_A$ all contain dams despite the vast number of alternative possibilities within the ImageNet classes which highlight memorization in $\text{SSL}_A$ between this specific patch of water and the dam.

**Visualizing Memorization Beyond Class Label.** Figure 7 shows four examples of badgers from the memorized set. In all four images, the periphery crop (pink square) does not contain any indication that the foreground object is a badger. Despite this, the KNN set using $\text{SSL}_A$ consistently produce images of badgers, while the same does not hold for $\text{SSL}_B$. More interestingly, the KNN using $\text{SSL}_A$ in Figure 7a all contain *European* badgers, while reconstructions in Figure 7b all contain *American* badgers, accurately reflecting the species of badger present in the respective training images. Since ImageNet-1K does *not* differentiate between these two species of badgers, our reconstructions show that SSL models can memorize information that is highly specific to a training sample beyond its class label[9].

# 6 Mitigation of *déjà vu* memorization

We cannot yet make claims on why *déjà vu* occurs so strongly for some SSL training settings and not for others. To gain some intuition for future work, we present additional observations that shed light on which parameters have the most salient impact on *déjà vu* memorization.

**Déjà vu memorization worsens by increasing number of training epochs.** Figure 4a shows how *déjà vu* memorization changes with number of training epochs for VICReg. The training set size is fixed to 300K samples. From 250 to 1000 epochs, the *déjà vu* score (red curve) grows *threefold*: from under 10% to over 30%. Remarkably, this trend in memorization is *not* reflected by the linear probe gap (dark blue), which only changes by a few percent beyond 250 epochs.

**Training set size has minimal effect on *déjà vu* memorization.** Figure 4b shows how *déjà vu* memorization responds to the model's training set size. The number of training epochs is fixed to 1000. Interestingly, training set size appears to have almost *no* influence on the *déjà vu* score (red line), indicating that memorization is equally prevalent with a 100K dataset and a 500K dataset. This result suggests that *déjà vu* memorization may be detectable even for large datasets. Meanwhile, the

---

[9]See Appendix A.1 for additional visualization experiments.

standard linear probe train-test accuracy gap *declines* by more than half as the dataset size grows, failing to represent the memorization quantified by our test.

**Training loss hyper-parameter has a strong effect.** Loss hyper-parameters, like VICReg's invariance coefficient (Figure 8a) or SimCLR's temperature parameter (Appendix Figure 14a) significantly impact *déjà vu* with minimal impact on the linear probe validation accuracy.

**Some SSL criteria promote stronger *déjà vu* memorization.** Table 9b demonstrates that the degree of memorization varies widely for different training criteria. VICReg and Barlow Twins have the highest *déjà vu* scores while SimCLR and Byol have the lowest. With the exception of Byol, all SSL models have more *déjà vu* memorization than the supervised model. Interestingly, different criteria can lead to similar linear probe validation accuracy and very different degrees of *déjà vu* as seen with SimCLR and Barlow Twins. Note that low degrees of *déjà vu* can still risk training image reconstruction, as exemplified by the SimCLR reconstructions in Figures 6 and 11.

**Larger models have increased *déjà vu* memorization.** Figure 9a validates the common intuition that lower capacity architectures (Resnet18/34) result in less memorization than their high capacity counterparts (Resnet50/101). We see the same trend for vision transformers as well.

**Guillotine regularization can help reduce *déjà vu* memorization.** Previous experiments were done using the projector embedding. In Figure 8b, we present how Guillotine regularization[3] (removing final layers in a trained SSL model) impacts *déjà vu* with VICReg[10]. Using the backbone embedding instead of the projector embedding seems to be the most straightforward way to mitigate *déjà vu* memorization. However, as demonstrated in Appendix A.7, backbone representation with low *déjà vu* score can still be leveraged to reconstruct some of the training images.

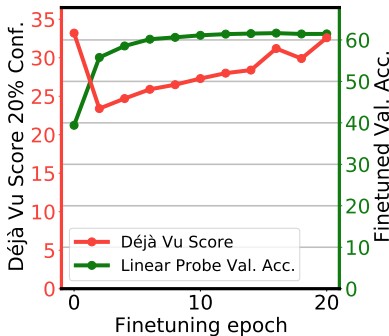

**Figure 10:** *Déjà vu* score when fine-tuning a pretrained VICReg model for 20 epochs. We observe that by fine-tuning we significantly increase the classification performances.

**A little bit of fine-tuning might help to reduce memorization** A common strategy in SSL is to fine-tune the model to solve the downstream task. In Figure 10, we show how the Dejavu score changes when fine-tuning a pretrained VICReg model. This pretrained model was trained on the set $\mathcal{A}$ for 1000 epochs and fine-tuned on a classification task on the set $\mathcal{A}$ for 20 epochs (which can be seen in the x axis on the figure). Interestingly, the DejaVu score decreases significantly in the first finetuning epochs while the validation accuracy is increasing. However after 5 epochs, the DejaVu score is increasing and after 20 epochs become almost as high at the original value before fine-tuning. This behavior indicates that even fine-tuning might not help in reducing DejaVu memorization.

## 7 Conclusion

We defined and analyzed *déjà vu* memorization, a notion of unintended memorization of partial information in image data. As shown in Sections 4 and 5, SSL models can largely exhibit *déjà vu* memorization on their training data, and this memorization signal can be extracted to infer or visualize image-specific information. Since SSL models are becoming increasingly widespread as foundation models for image data, negative consequences of *déjà vu* memorization can have profound downstream impact and thus deserves further attention. Future work should focus on understanding how *déjà vu* emerges in the training of SSL models and why methods like Byol are much more robust to *déjà vu* than VICReg and Barlow Twins. In addition, trying to characterize which data points are the most at risk of *déjà vu* could be crucial to get a better understanding on this phenomenon.

---

[10]Further experiments are available in Appendix A.7.

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

# A    Appendix

## A.1    Additional reconstruction examples

The two reconstruction experiments of Section 5 are each exemplified within one class. However, we see strong reconstructions using $\text{SSL}_A$ in several classes, and similar experimental results. To demonstrate this, we repeat the experiment of Section 5 using the *yellow garden spider* class and the experiment of 5 using the *aircraft carrier* class.

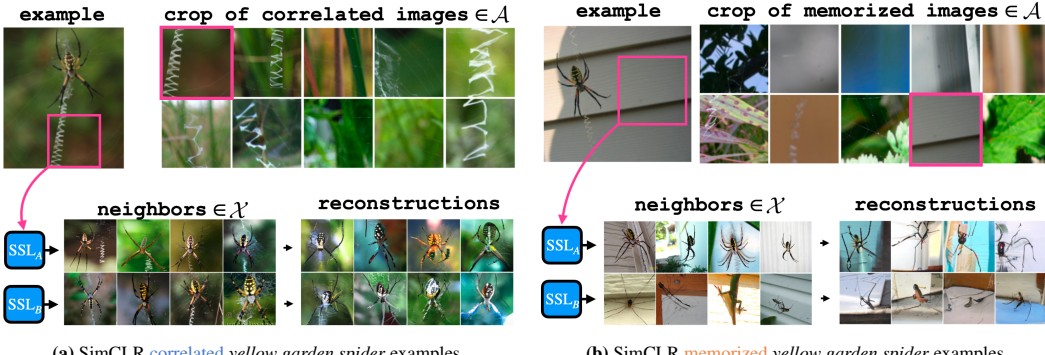

**(a)** SimCLR correlated *yellow garden spider* examples          **(b)** SimCLR memorized *yellow garden spider* examples

**Figure 11:** Visualizing the distinction between *déjà vu* memorization and correlation in the *yellow garden spider* class. Left, we see the periphery crops of the ten 'most correlated' images: those where both $\text{KNN}_A$ and $\text{KNN}_B$ have high confidence. Seven of these crops clearly depict a stabilimentum: the signature zig-zag web pattern sewn by spiders of the *argiope* genus, thus revealing the concealed spider by correlation. Right, we see the periphery crops of the ten 'most memorized' images: those that have the highest confidence discrepancy between $\text{KNN}_A$ (high confidence) and $\text{KNN}_B$ (low confidence). Nearly all of these crops show generic blurred views of the background with no evidence of the foreground spider. Below, we show the public set nearest neighbors of the pink highlighted crop. We see that the target model ($\text{SSL}_A$) can be used to find back the yellow garden spider spider in both the memorized and correlated cases. The reference model ($\text{SSL}_B$) can only be used to find back this type of spider in the correlated case.

**Selection of Memorized and Correlated Images:**    The images of Figure 6 and 11 were chosen methodically as follows.

*Image selection:* The 20 images of Figures 6 and 11 are selected deterministically using label inference accuracy and KNN confidence score. The 10 most correlated images are those images in the correlated set (both models infer label correctly) of $\mathcal{A}$ with the highest confidence agreement between models $\text{SSL}_A$ and $\text{SSL}_B$. To measure confidence agreement we take the minimum confidence of the two models. The 10 most memorized images are those images in the memorized set (only target model infers the label correctly) of $\mathcal{A}$ with the highest confidence difference between models $\text{SSL}_A$ and $\text{SSL}_B$.

*Class selection:* To find classes with a high degree of *déjà vu* , classes were sorted by the label inference accuracy gap between the target and reference model. We selected the class based on a handful of criteria. First, we prioritized classes without images of human faces, thereby removing classes like 'basketball', 'bobsled', 'train station', and even 'tench' which is a fish often depicted in the hands of a fisherman. Second, we prioritized classes that include at least ten images with a high confidence difference between the target and reference models ('most memorized' images described above) and at least ten images with high confidence agreement ('most correlated' images described above). This led us to the *dam* and *yellow garden spider* classes.

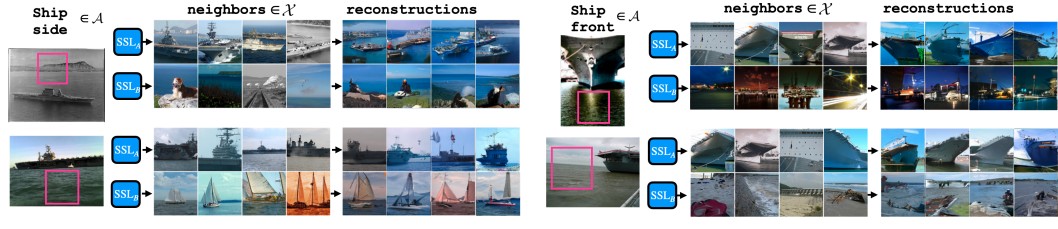

**(a)** extracting side of ship from VICReg  **(b)** extracting front of ship from VICReg

**Figure 12:** Visualization of *déjà vu* memorization beyond class label. Both $SSL_A$ and $SSL_B$ are VICReg models. The four images shown belong to the memorized set of $SSL_A$ from the *aircraft carrier* class. KNN visualization using embeddings from $SSL_A$ can reveal not only the correct class label, but also the orientation of the ship: the side of the ship (left) and the front of the ship (right) given only a generic crop of the background sky and/or water. Such information does not appear to be memorized by the reference model $SSL_B$.

**Selection of Beyond-Label-Inference Images:**  The images of Figure 7 and 12 were chosen methodically as follows.

*Image selection:* The four images of Figures 7 and 12 are selected using KNN confidence score, and, necessarily, hand picked selection for unlabeled features. Within a given class, we look at the top 40 images with highest target model KNN confidence scores. We then filter through these images to identify a distinguishable feature like different species within the same class or different object positions within the same class. This step is necessary because we are looking for features that are not labeled by ImageNet. We then choose two of these top 40 with one feature (e.g. American badger) and two with the alternative feature (e.g. European badger).

*Class selection:* To find classes with a high degree of *déjà vu* , classes were sorted by the target model's top-40 KNN confidence values within each class. As in the memorization vs. correlation experiment, we prioritized classes without images of human faces.

### A.2 Details on the experimental setup

#### A.2.1 Details on dataset splits

There are 1,281,167 training images in the ImageNet dataset. Within these images, only 456,567 of them have bounding boxes annotations (which are needed to compute the Deja Vu score). The private set $\mathcal{A}$ and $\mathcal{B}$ are sampled from these 456,567 bounding boxes annotated images in such a way that set $\mathcal{A}$ and $\mathcal{B}$ are disjoint.

If we remove the 456,567 bounding boxes annotated images from the 1,281,167 training images, we get 824,600 remaining images without annotations which never overlap with $\mathcal{A}$ or $\mathcal{B}$. From this set of 800K images, we took 500k images as our public set $X$. So now, we have three non overlapping sets $\mathcal{A}$, $\mathcal{B}$, and X. Then, if we remove the 500K public set images from the 824,600 images without annotations, it leaves us with 324,600 images that are neither in $\mathcal{A}$, $\mathcal{B}$ or $X$. For simplicity, let us call this set of remaining 324,600 images the set $\mathcal{C}$. Then, we have split the entire ImageNet training set into four non-overlapping splits called $\mathcal{A}$, $\mathcal{B}$, $\mathcal{C}$ and $X$.

When running our experiments with a small number of training images, we only use the set $\mathcal{A}$ to train $\text{SSL}_A$ and the set $\mathcal{B}$ to train $\text{SSL}_B$ and then use the set $X$ as a public set for evaluation. However, to run larger scale experiments, we use as additional training data for $\text{SSL}_A$ and $\text{SSL}_B$: the images sampled from the set $\mathcal{C}$. Here, $\text{SSL}_A$ will still be trained on set $\mathcal{A}$ but it will be augmented with images from set $\mathcal{C}$. The same goes for $\text{SSL}_B$ which will still be trained on the set $\mathcal{B}$ but augmented with images from the set $\mathcal{C}$. As such, some images sampled from $\mathcal{C}$ to train $\text{SSL}_A$ or to train $\text{SSL}_B$ might overlap. However, this is not an issue since the evaluation is done using only images from the bounding boxes annotated set $\mathcal{A}$ and set $\mathcal{B}$ which are never overlapping.

To identify memorization, our tests only attempt to infer the labels of the unique examples $\overline{\mathcal{A}}$ and $\overline{\mathcal{B}}$ that differentiate the two private sets. The periphery crop, $\text{crop}(A_i)$, is computed as the largest possible crop that does not intersect with the foreground object bounding box. In some instances the largest periphery crop is small, and not high enough resolution to get a meaningful embedding. To circumvent this, we only run the test on bounding box examples where the periphery crop is at least $100 \times 100$ pixels.

Each size of training set, 100k to 500k, includes an equal number of examples per class in both sets $\mathcal{A}$ and $\mathcal{B}$. The total bounding box annotated examples of each class are evenly divided between $\overline{\mathcal{A}}$ and $\overline{\mathcal{B}}$. The remaining examples in each class are the examples from $\mathcal{C}$. We reiterate that the bounding box examples in set $\mathcal{A}$ are *unique* to set $\mathcal{A}$, and thus can only be memorized by $\text{SSL}_A$.

The disjoint public set, $X$, contains ground truth labels but no bounding-box annotations. The size and content of $X$ remains fixed for all tests.

#### A.2.2 Details on the training setup

**Model Training:** We use PyTorch [23] with FFCV-SSL [5]. All models are trained for 1000 epochs with model checkpoints taken at 50, 100, 250, 500, 750, and 1000 epochs. We note that 1000 epochs is used in the original papers of both VICReg and SimCLR. All sweeps of epochs use the 300k dataset. All sweeps of datasets use the final, 1000 epoch checkpoint. We use a batch size of 1024, and LARS optimizer [31] for all SSL models. All models use Resnet101 for the backbone. As seen in Appendix A.6, a Resnet50 backbone results in *déjà vu* consistent with that of Resnet101.

**VICReg Training:** VICReg is trained with the 3-layer fully connected projector used in the original paper with layer dimensions 8192-8192-8192. The invariance, variance, and covariance parameters are set to $\lambda = 25, \mu = 25, \nu = 1$, respectively, which are used in the original paper [2]. The LARS base learning rate is set to 0.2, and weight decay is set to 1e-6.

**SimCLR Training:** SimCLR is trained with the 2-layer fully connected projector used in the original paper with layer dimensions 2048-256. The temperature parameter is set to $\tau = 0.15$. The LARS base learning rate is set to 0.3, and weight decay is set to 1e-6.

**Supervised Training:** Unlike the SSL models, the supervised model is trained with label access using cross-entropy loss. To keep architectures as similar as possible, the supervised model also uses a Resnet101 backbone and the same projector as VICReg. A final batchnorm, ReLU, and linear layer

is added to bring the 8192 dimension projector output to 1000-way classification activations. We use these activations as the supervised model's projector embedding. The supervised model uses the LARS optimizer with learning rate 0.2.

### A.2.3 Details on the evaluation setup

**KNN:** For each test, we build two KNN's: one using the target model, $\text{SSL}_A$ (or $\text{CLF}_A$), and one using the reference model $\text{SSL}_B$ (or $\text{CLF}_B$). As depicted in Figure 2, each KNN is built using the projector embeddings of all images in the public set $\mathcal{X}$ as the neighbor set. When testing for memorization on an image $A_i \in \mathcal{A}$, we first embed $\text{crop}(A_i)$ using $\text{SSL}_A$, and find its $K = 100$ $L_2$ nearest neighbors within the $\text{SSL}_A$ embeddings of $\mathcal{X}$. See section A.4 for a discussion on selection of $K$. We then take the majority vote of the neighbors' labels to determine the class of $A_i$. This entire pipleine is repeated using reference model $\text{SSL}_B$ and its KNN to compute reference model accuracy.

In practice, all of our quantitative tests are repeated once with $\text{SSL}_A$ as the target model (recovering labels of images in set $\mathcal{A}$) and again with $\text{SSL}_B$ as the target model (recovering labels of images in set $\mathcal{B}$). All results shown are the average of these two tests. Throughout the paper, we describe $\text{SSL}_A$ as the target model and $\text{SSL}_B$ as the reference model for ease of exposition.

**RCDM:** The RCDM is trained on a face-blurred version of ImageNet [13] and is used to decode the SSL backbone embedding of an image back into an approximation of the original image. All RCDMs are trained on the public set of images $\mathcal{X}$ used for the KNN. A separate RCDM must be trained for each SSL model, since each model has a unique mapping from image space to embedding space.

At inference time, the RCDM is used to reconstruct the foreground object given only the periphery cropping. To produce this reconstruciton, the RCDM needs an approximation of the backbone embedding of the original image. The backbone of image $A_i$ is approximated by **1)** computing crop embedding $\text{SSL}_A^{\text{proj}}(\text{crop}(A_i))$, **2)** finding the five public set nearest neighbors of the crop embedding, and **3)** averaging the five nearest neighbors' backbone embeddings. In practice, these public set nearest neighbors are often a very good approximation of the original image, capturing aspects like object class, position, subspecies, etc..

## A.3 Sample-level memorization

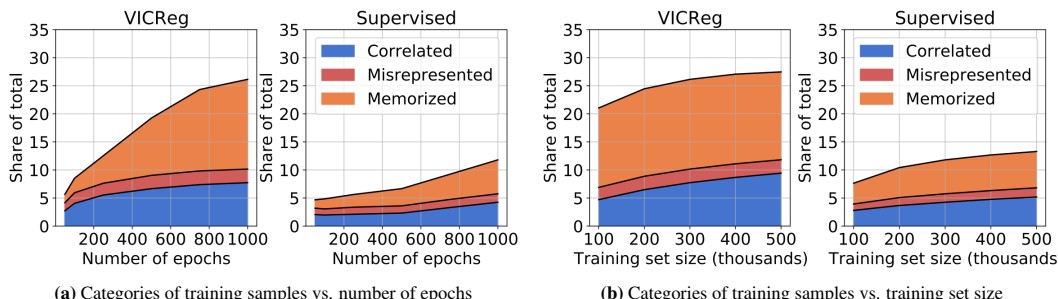

**(a)** Categories of training samples vs. number of epochs

**(b)** Categories of training samples vs. training set size

**Figure 13:** Partition of samples $A_i \in \mathcal{A}$ into the four categories: unassociated (not shown), memorized, misrepresented and correlated. The memorized samples—ones whose labels are predicted by $\text{KNN}_A$ but not by $\text{KNN}_B$—occupy a significantly larger share for VICReg compared to the supervised model, indicating that sample-level *déjà vu* memorization is more prevalent in VICReg.

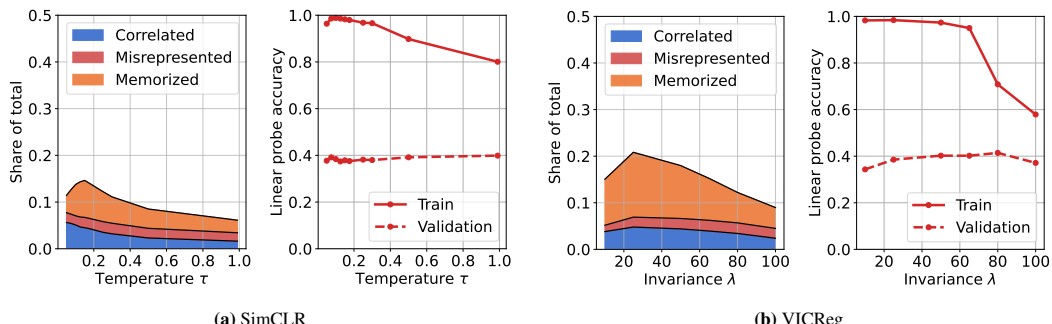

**(a)** SimCLR

**(b)** VICReg

**Figure 14:** Effect of SSL hyperparameter on *déjà vu* memorization. The left plot of Figures 14a and 14b show the size of the memorized set as a function of the temperature parameter for SimCLR and invariance parameter for VICReg, respectively. *Déjà vu* memorization is the highest within a narrow band of hyperparameters, and one can mitigate against *déjà vu* memorization by selecting hyperparameters outside of this band. Doing so has negligible effect on the quality of SSL embeddings as indicated by the linear probe accuracy on ImageNet validation set.

Many SSL algorithms contain hyperparameters that control how similar the embeddings of different views should be in the training objective. We show that these hyperparameters directly affect *déjà vu* memorization. Figure 14 shows the size of the memorized set for SimCLR (left) and VICReg (right) as a function of their respective hyperparameters, $\tau$ and $\lambda$. We observe that the memorized set is largest within a relatively narrow band of hyperparameter values, indicating strong *déjà vu* memorization. By selecting hyperparameters outside this band, *déjà vu* memorization sharply decreases while the linear probe validation accuracy on ImageNet remains roughly the same.

## A.4 Selection of $K$ for KNN

In this section, we describe the impact of $K$ on the KNN label inference accuracy.

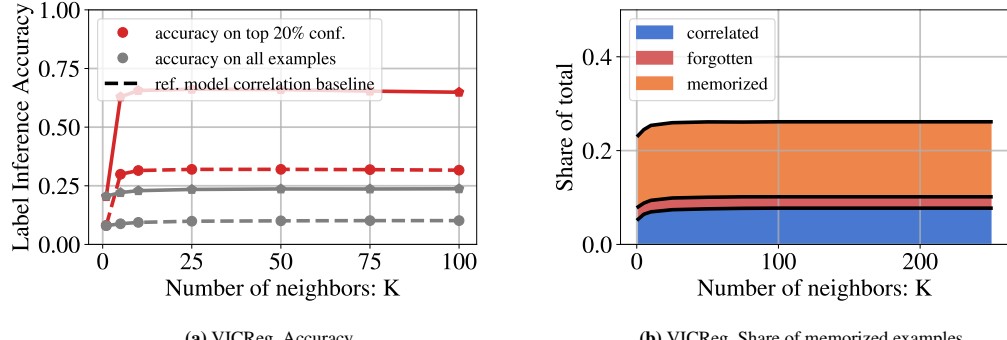

**(a)** VICReg, Accuracy

**(b)** VICReg, Share of memorized examples

**Figure 15:** Impact of $K$ on label inference accuracy for target and reference models. **Left:** the population-level label inference accuracy experiment of Section 4.1 on VICReg vs. $K$. **Right:** the individualized memorization test of Section 4.2 on VICReg vs. $K$. In both cases, we see that our tests are relatively robust to choice of $K$ beyond $K = 50$.

Figure 15 above shows how the tests of Section 4 change with number of public set nearest neighbors $K$ used to make label inferences. Both tests are relatively robust to any choice of $K$. Results are shown on VICReg trained for 1k epochs on the 300k dataset. We see that any choice of $K$ greater than 50 and less than the number of examples per class (300, in this case) appears to have good performance. Since our smallest dataset has 100 images per class, we chose to set $K = 100$ for all experiments.

## A.5 Effect of SSL criteria

We repeat the quantitative memorization tests of Section 4 on different models: VICReg[2], Barlow-Twins[33], Dino[10], Byol[18], SimCLR[34] and a supervised model in Figure 16. We observe differences between SSL training criteria with respect to *déjà vu* memorization. The easy ones to attack are VICReg and Barlow Twins whereas SimCLR and Byol are more robust to these attacks. While the degree of memorization appears to be reduced for SimCLR compared with VICReg, it is still stronger than the supervised baseline.

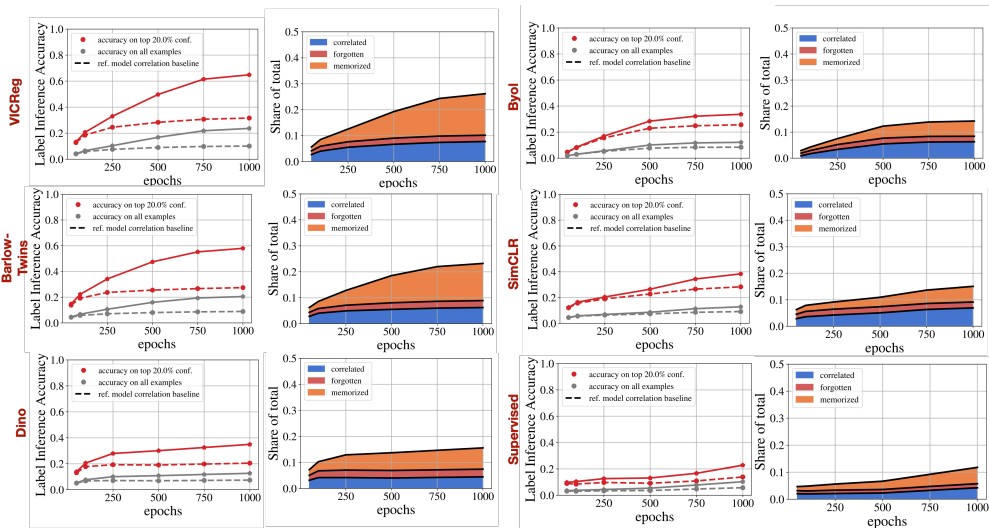

**Figure 16:** Comparison of *déjà vu* memorization for VICReg, Barlow Twins, Dino, Byol, SimCLR, and a supervised model. All tests are described in Section 4. We are showing *déjà vu* vs. number of training epochs. We see that SimCLR (center row) shows less *déjà vu* than VICReg, yet marginally more than the supervised model. Even with this reduced degree of memorization, we are able to produce detailed reconstructions of training set images, as shown in Figures 6 and 11.

## A.6 Effect of Model Architecture and Complexity

Results shown in the main paper use Resnet101 for the model backbone. To understand the relationship between *déjà vu* and overparameterization, we compare with the smaller Resnet50 and Resnet18 in Figure 17. Overall, we find that increasing the number of parameters of the model leads to higher degree of *déjà vu* memorization. The same trend holds when using Vision Transformers (VIT-Tiny, -Small, -Base, and -Large with patch size of 16) of various sizes as the SSL backbone, instead of a Resnet. This highlights that *déjà vu* memorization is not unique to convolution architectures.

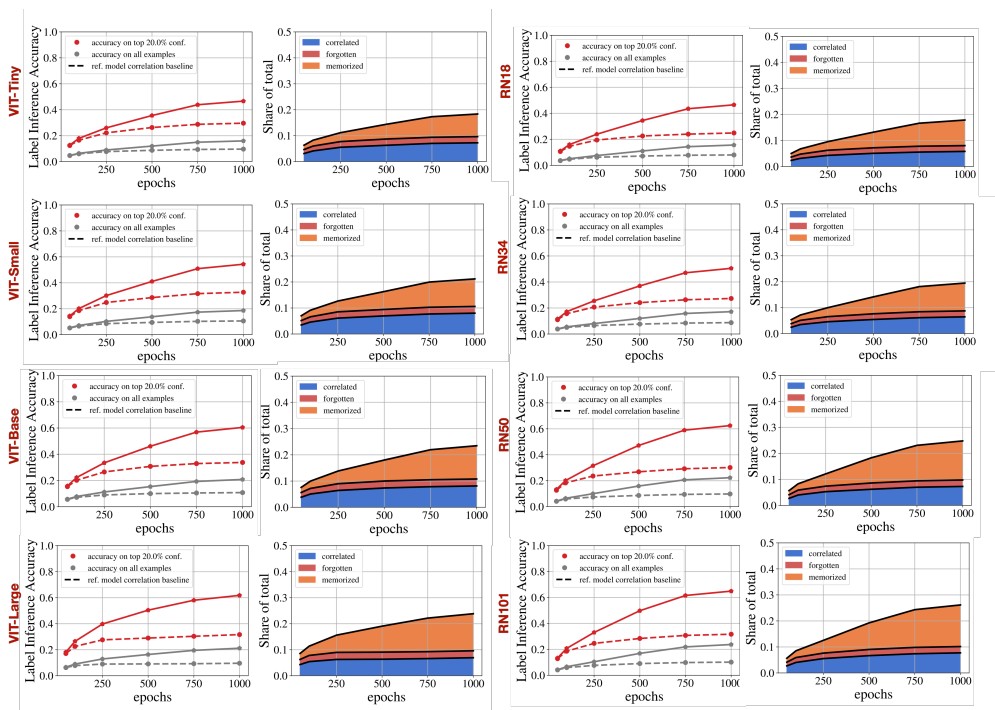

**Figure 17:** Comparison of VICReg *déjà vu* memorization for different architectures and model sizes. On the left, we present deja vu memorization using VIT architectures (from vit-tiny in the first row to vit-base in the last row). On the right, we use Resnet based architectures (from resnet18 in the first row to resnet101 in the last row). All tests are described in Section 4, with the plots showing *déjà vu* vs. number of training epochs. Reducing model complexity from Resnet101 to Resnet18 or from Vit-Large to Vit-tiny has a significant impact on the degree of memorization.

## A.7 The impact of Guillotine Regularization on Deja Vu

In our experiments, we show *déjà vu* using the projector representation. The SSL loss directly incentivizes the projector representation to be invariant to random crops of a particular image. As such, we expect the projector to be the *most* overfit and produce the strongest *déjà vu* . Here, we study whether earlier representations between the projector and backbone exhibit less *déjà vu* memorization. This phenomenon – 'guillotine regularization' – has recently been studied from the perspective of generalization in Bordes et al. [3]. Here, we study it from the perspective of *memorization*.

To show how guillotine regularization impacts *déjà vu* , we repeat the tests of Section 4 on each layer of the VICReg projector: the 2048-dimension backbone (layer 0) up to the projector output (layer 3). We evaluate whether memorization is indeed reduced for the more *regularized* layers between the projector output and the backbone.

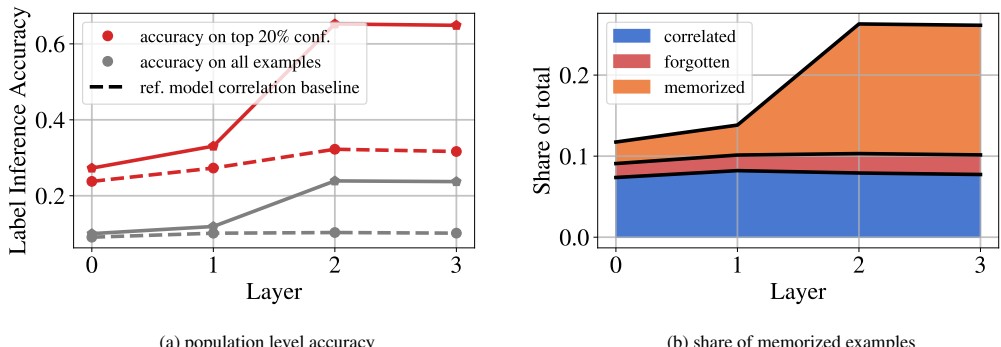

(a) population level accuracy

(b) share of memorized examples

**Figure 18:** *déjà vu* memorization versus layer from backbone (0) to projector output (3). The memorization tests of Section 4 are evaluated at each level of the VICReg projector. We see that *déjà vu* is significantly stronger closer to the projector output and nearly zero near the backbone. Interestingly, most memorization appears to occur in the final two layers of VICReg.

Figure 18 shows how guillotine regularization significantly reduces the degree of memorization in VICReg. The vast majority of VICReg's *déjà vu* appears to occur in the final two layers of the projector (2,3): in earlier layers (0,1), the label inference accuracy of the target model and reference model are comparable. This suggests that – like the hyperparameter selection of Section 6 – guillotine regularization can also significantly mitigate *déjà vu* memorization. In the following, we extend this result to SimCLR and supervised models by measuring the degree of *déjà vu* in the backbone (layer 0) versus training epochs and dataset size.

**Comparison of *déjà vu* in projector and backbone vs. epochs and dataset size** Since the backbone is mostly used at inference time, we now evaluate how much *déjà vu* exists in the backbone representation for VICReg and SimCLR. We repeat the tests of Section 4 versus training epochs and train set size.

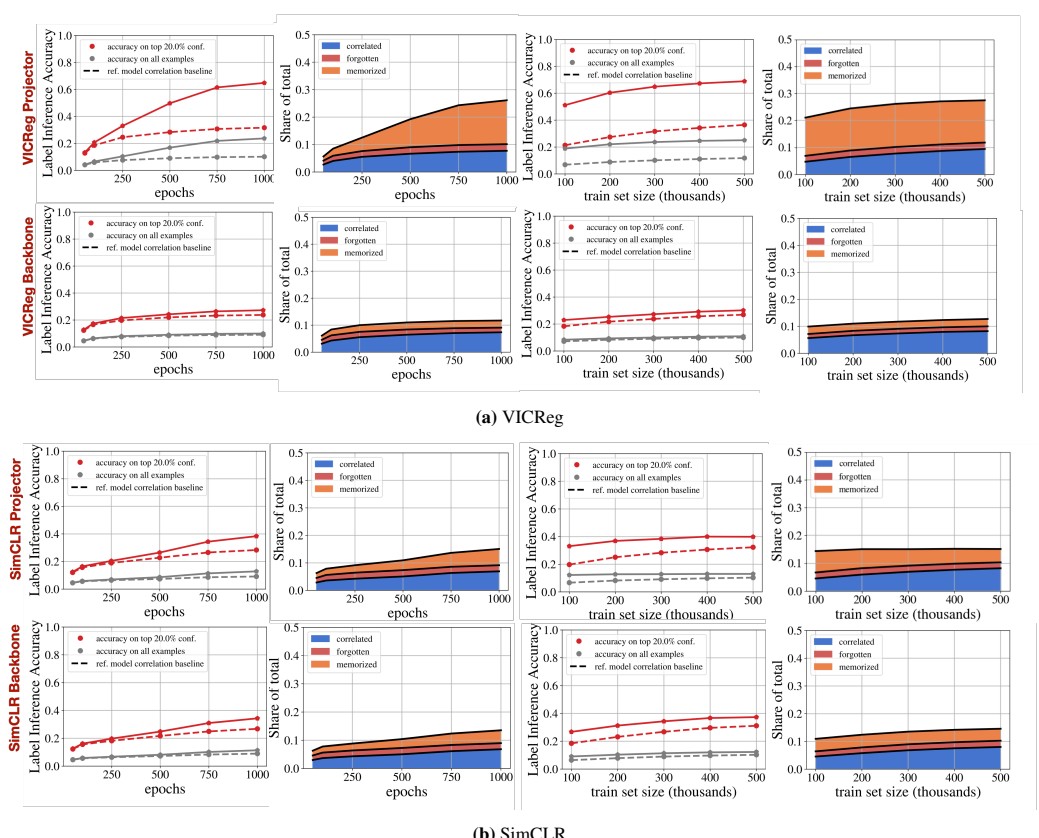

**(a)** VICReg

**(b)** SimCLR

**Figure 19:** Accuracy of label inference on VICReg and SimCLR using projector and backbone representations. **First two columns:** Effect of training epochs on memorization for each representation. **Last two columns:** Effect of training set size on memorization for each representation. In contrast with VICReg, the *déjà vu* memorization detected in SimCLR's projector and backbone representations is quite similar. While SimCLR's projector memorization appears weaker than that of VICReg, its backbone memorization is markedly stronger. This kind be easily explained as a byproduct of Guillotine Regularization [3], i.e. removing layers close to the objective reduce the bias of the network with respect to the training task. Since SimCLR's projector has fewer layers than VICReg's, the impact of Guillotine Regularization is less salient.

Figure 19 shows that, indeed, *déjà vu* is significantly reduced in the backbone representation. For SimCLR, however, we see that backbone memorization is comparable with projector memorization. In light of the Guillotine regularization results above, this makes some sense since SimCLR uses fewer layers in its projector. Given that we were able to generate accurate reconstructions with the SimCLR projector (see Figures 11 and 6), we now evaluate whether we can produce accurate reconstructions of training examples using the SimCLR backbone alone.

**Reconstructions using SimCLR Backbone Only:** The above label inference results show that the SimCLR backbone exhibits a similar degree of *déjà vu* memorization as the projector does. To evaluate the risk of such memorization, we repeat the reconstruction experiment of Section 5 on the *dam* class using the SimCLR backbone instead of its projector.

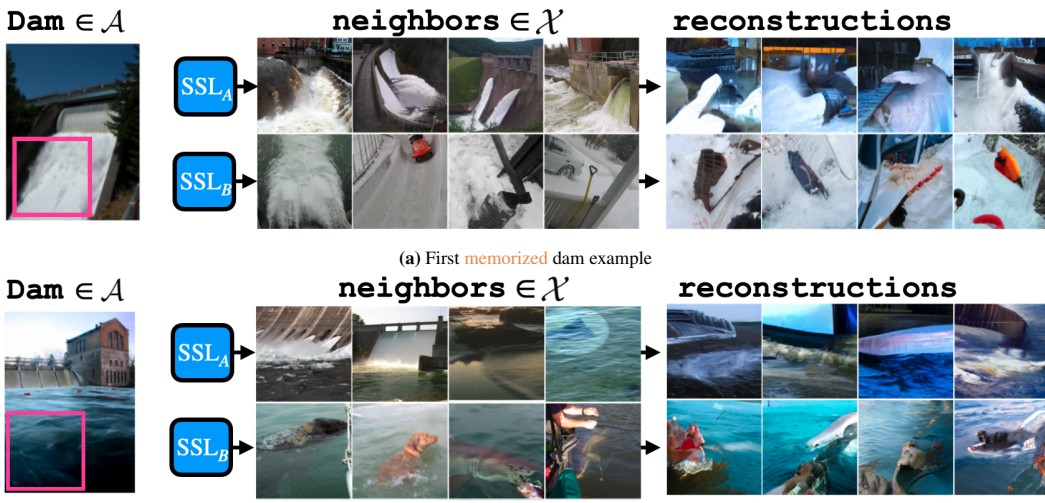

(a) First memorized dam example

(b) Second memorized dam example

**Figure 20:** Instances of *déjà vu* memorization by the SimCLR backbone representation. Here, the backbone embedding of the crop is used instead of the projector embedding on the same training images used in Figure 6. Interestingly, we see that *déjà vu* memorization is still present in the SimCLR backbone representation. Here, the nearest neighbor set recovers dam given an uninformative crop of still or running water. Even without projector access, we are able to reconstruct images in set $\mathcal{A}$ using $\text{SSL}_A$, and are unable using $\text{SSL}_B$.

Figure 20 demonstrates that we are able to reconstruct training set images using the SimCLR backbone alone. This indicates that *déjà vu* memorization can be leveraged to make detailed inferences about training set images without *any* access to the projector. As such, withholding the projector for model release may not be a strong enough mitigation against *déjà vu* memorization.

## A.8 Detecting *Déjà vu* without Bounding Box Annotations

The memorization tests presented critically depend on bounding box annotations in order to separate the foreground object from the periphery crop. Since such annotations are often not available, we propose a heuristic test that simply uses the lower left corner of an image as a surrogate for the periphery crop. Since foreground objects tend to be near the center of the image, the corner crop usually excludes the foreground object and does not require a bounding box annotation.

Figure 21 demonstrates that this heuristic test can successfully capture the trends of the original tests (seen in Figure 16) *without* access to bounding box annotations. However, as compared to Figure 16, the heuristic tends to slightly underestimate the degree of memorization. This is likely due to the fact that some corner crops partially include the foreground object, thus enabling the KNN to successfully recover the label with the reference model where it would have failed with a proper periphery crop that excludes the foreground object.

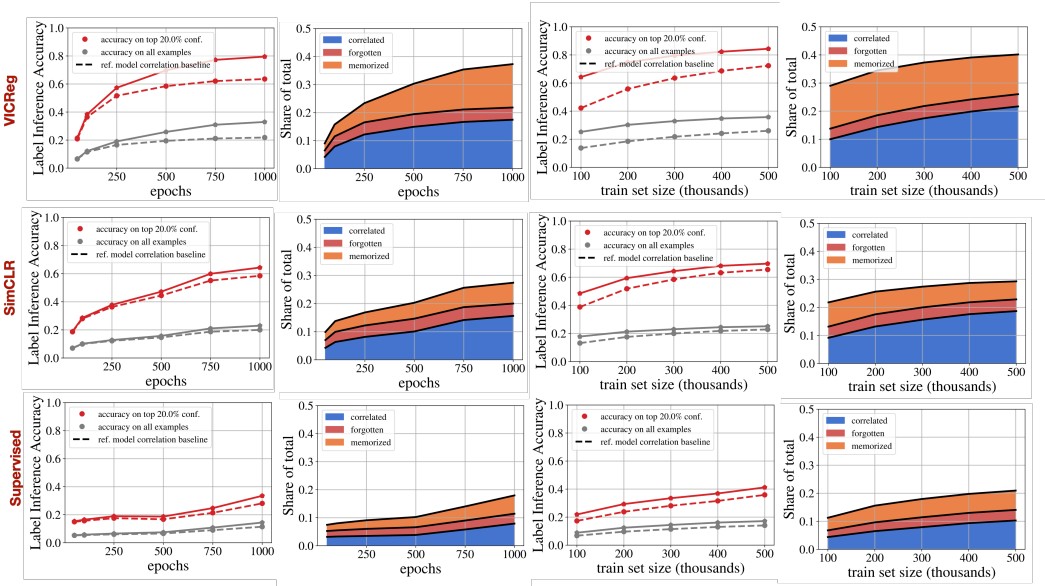

**Figure 21:** *Déjà vu* memorization using a simple corner crop instead of the periphery crop extracted using bounding box annotations. While the heuristic overall underestimates the degree of *déjà vu*, it roughly follows the same trends versus dataset size and training epochs. This is crucial, since it allows us to estimate *déjà vu* without access to bounding box annotations.

