# OpenReview forum: "Do SSL Models Have Déjà Vu? A Case of Unintended Memorization in Self-supervised Learning"
_NeurIPS.cc/2023/Conference — NeurIPS 2023 poster_

### Official Review · Reviewer_mdS9 · 2023-07-03

**Soundness:** 3 good
**Presentation:** 3 good
**Contribution:** 3 good
**Rating:** 6
**Confidence:** 3

**Summary:**

This paper thoroughly studies a particular undesired aspect of state-of-the-art self-supervised learning models: deja-vu memorization, where the models end up memorizing specific parts of the image instead of learning just a semantically meaningful association. The authors provide extensive empirical results to show when SSL models might memorize, and what aspects of training might minimize this effect.

**Strengths:**

* The authors introduce novel idea and testing methodology for SSL models
* Provides comprehensive empirical results on different aspects of SSL modeling and training and memorization
* Additional results on ablations and using different representations and aspects of the setup help understand the author’s proposed method better
* Clear presentation of results and setup.

**Weaknesses:**

* Authors do not seem to talk about the impact of different augmentations on memorization, if any. Augmentations are very important for SSL performance, maybe they impact memorization?
* It is hard to judge how confidently we can estimate the memorization if deploying. We compare to only one other reference SSL model.
* The importance of using the generative model is not quite clear. From the result figures, the kNN images themselves seem to do enough.
* Only shows using ImageNet data. No way to know if it will generalize to other datasets as well.

**Questions:**

* How do different augmentations used by SSLs impact memorization?
* Is using just one other SSL model enough to give confident results of memorization? Should we include more reference SSL models to increase confidence?
* How many samples would be needed in the X public dataset? Could be helpful to understand the scale if deployed.
* When is the generative model actually needed, when is it necessary beyond just looking at the k nearest neighbors?

**Limitations:**

* The lack of a way to estimate the confidence or uncertainty in the memorization score.
* Although shown for many models, the results are shown only for a single dataset. Using more datasets could have been nice, maybe using CelebA to show background info is memorized to learn person’s identity.

---

> ### Author Rebuttal · Authors · 2023-08-04
>
> Thank you for reading our work, and for your suggestions/questions! We have attempted to answer your primary questions and concerns below. Please let us know if you have any further questions, or if anything is unclear.
>
> **Authors do not seem to talk about the impact of different augmentations on memorization, if any. Augmentations are very important for SSL performance, maybe they impact memorization?**
>
> In our instance, the most important augmentation is the Random Cropping one. If we don’t use random crop, there is no way for the network to learn to associate a specific patch to a given image. In fact, we also show in the paper some memorization occurring with a classifier which is trained only with random cropping. Concerning SSL, it is true that additional color based augmentation like color-jitter are used which make the task of mapping the representation of different crops much harder.
>
> As request, we ran experiments in which we change the SSL data augmentations. For the first one we remove the color-jitter augmentation from the SSL augmentations (leaving cropping, grayscale, solarization and blur), while for the second one we kept only cropping as data augmentation.
>
> | | All Augmentations | Without ColorJitter | Only Cropping |
> |---|---|---|---|
> | KNN Accuracy | 40.6 | 26.6 | 18.2 |
> | DejaVu Score | 33.2 | 24.9 | 12,9 |
>
> ---
> As you can see, removing augmentations decreases as expected the KNN accuracy of the models. In consequence, It also reduces the deja vu score. The issue is that it is not really possible to disentangle augmentation from dejavu memorization from generalization. SSL models needs strong augmentations and if we remove them, we just learn bad representation that cannot be leveraged for any downstream tasks.
>
> **It is hard to judge how confidently we can estimate the memorization if deploying. We compare to only one other reference SSL model.**
>
> Our method measures model $A$’s memorization of set $A$ and model $B$’s memorization of set $B$, and reports the average. We consistently find that the degree of memorization of these two disjoint sets are nearly equal. Given that both model $A$ and $B$ appear to show systematic memorization of their disjoint training examples, we believe that this confidently reveals the existence of memorization in training.
>
> To show memorization of a single example, we agree that one would need to train several reference models (to e.g. confirm that the reference model rarely guesses a foreground Swan given a water background). However, this would significantly increase the computational demand of the test. We feel that our test confidently captures the existence of memorization without untenable computational demands.
>
> **The importance of using the generative model is not quite clear. From the result figures, the kNN images themselves seem to do enough.**
>
> Thank you for your feedback on this point. The generative model offers a single reconstruction by averaging the kNN examples. It is helpful to know that this did not add much to your read of the paper. We will edit the paper to de-emphasize the generative model reconstructions.
>
> **[...] only shows using ImageNet data. No way to know if it will generalize to other datasets as well.**
>
> Our choice of a curated dataset like ImageNet was intentional for two reasons.
>
> First, ImageNet comes with bounding box annotations that allow us to precisely separate the background (e.g. water)  from the labeled foreground object (e.g. swan), allowing us to run our quantitative memorization tests. When bounding boxes are not available, we propose a heuristic test (see Appendix A.5) that simply takes a corner crop of the image which most likely removes the foreground object. We show that this heuristic offers a good lower bound of memorization.
>
> Second, ImageNet is largely deduplicated allowing us to show memorization that cannot be fixed by better dataset curation. Prior works on dataset reconstruction e.g. [1, 2] show near-exact memorization that mostly occurs with highly duplicated examples, and can be largely avoided with deduplication. We feel that this is a significant contribution of our work.
>
> **[...] maybe using CelebA to show background info is memorized to learn person’s identity.**
>
> We agree that such a study is important, however there are privacy and legal concerns in play here. Consent was never obtained from the people appearing in the CelebA dataset which might pose legal and ethical concern (and blurring the faces in CelebA will just make the dataset unusable). Using ImageNet, we have a large diversity of images which offers significantly more variety than smaller scale dataset like CelebA and others.
>
> [1] Carlini, Nicholas, Florian Tramer, Eric Wallace, Matthew Jagielski, Ariel Herbert-Voss, Katherine Lee, Adam Roberts, et al. “Extracting Training Data from Large Language Models.” ArXiv:2012.07805 [Cs], December 14, 2020. http://arxiv.org/abs/2012.07805.
>
> [2] Carlini, N., Hayes, J., Nasr, M., Jagielski, M., Sehwag, V., Tramer, F., ... & Wallace, E. (2023). Extracting training data from diffusion models. arXiv preprint arXiv:2301.13188.

---

> > ### Comment · Reviewer_mdS9 · 2023-08-11
> >
> > I thank the authors for answering my questions and doubts, especially regarding data augmentation. It makes perfect sense that fewer augmentations reduce the quality of the representations, but it is still interesting that it also facilitates memorization. I also perfectly understand the computing requirements for training the models and agree that results holding for $A$ to $B$ and $B$ to $A$ provide some initial confidence in the results.
> > While CelebA is a tricky dataset to use for works on privacy, it would indeed be interesting to see future results on some datasets where such privacy risks are definitively being leaked, e.g., in medical image data (maybe CheXpert?), and if a crop of an image (e.g., some hospital imaging device artifact) ends up leaking chest result data of individuals scanned by that machine. This is not to take away from the results shown.
> >
> > Finally, while the authors already show the impact of the SSL algorithm chosen and the data augmentations on Deja-vu, I am pretty interested to see future work on memorization in self-supervised methods and how other training choices impact Deja-vu, e.g., imposing adversarial robustness [1,2].
> >
> > [1] Kim, Minseon, Jihoon Tack, and Sung Ju Hwang. "Adversarial self-supervised contrastive learning." Advances in Neural Information Processing Systems 33 (2020): 2983-2994.
> >
> > [2] Chen, Tianlong, et al. "Adversarial robustness: From self-supervised pre-training to fine-tuning." Proceedings of the IEEE/CVF Conference on Computer Vision and Pattern Recognition. 2020.

---

### Official Review · Reviewer_H9Sq · 2023-07-04

**Soundness:** 4 excellent
**Presentation:** 3 good
**Contribution:** 3 good
**Rating:** 8
**Confidence:** 3

**Summary:**

This paper shows how self-supervised learning (SSL) models can memorize information about their training data in ways that are not intended. SSL is a technique for learning representations of unlabeled data by training a model to solve a "pretext" task; for example, we might take training images A, B, and C and crop them in different ways, producing sets of patches (A1,...,An), (B1,...,Bn), and (C1,...,Cn). The algorithm tries to find an embedding of these patches that makes Ai similar to Aj, while minimizing the similarity between Ai and Bj. These embeddings are used for a variety of downstream tasks.

This paper shows that the representations of training patches can contain surprising information about the training data. In the headlining example, the authors take an image of a black swan from the training data, extract a water-only patch, and produce an embedding of this patch. Then a generative model, which (ideally? see below) has no access to the training data, produces an image from this embedding: it contains a (different?) black swan. The water-only patch contained no information about swan, so the SSL model "remembered" what was in the image the patch came from.

The trick fails when the same patch is run through an SSL model that was trained on different data: it happens only because the patch was seen in the training data.

The paper has two methods for detecting such memorization. The first predicts the class label given the patch's representation. The second, mentioned above, visually inspects images generated from the representation. In both cases the classification/generation is done by a model trained on a different data set. The label-prediction method demonstrates quantitatively that the representation contains information about the label, while the visual inspection reveals finer details: the paper contains a great example of how SSL models can learn and memorize the distinction between European and American badgers, even though both share an ImageNet label.

**Strengths:**

If the authors resolve my dataset question below, I am happy to call this a very clean and original demonstration on a significant and relavant topic. It is definitely a paper to generate good discussion.

The paper performs extensive experiments, showing how the results change as they turn various knobs. The quantitative and qualitative techniques complement each other nicely.

The work lays out clear directions for future work.


**Weaknesses:**

As I understand from Appendices A.2.1 and A.2.3, "private" sets A and B come from ImageNet 1k, which is a subset of ImageNet. The "public" set X is a face-blurred version of ImageNet. This appears to contradict the experimental setup of Section 3.1 and, if it is the case, means I am not sure how to interpret the results. I could be mistaken and am open to raising my score. EDIT: the rebuttal adequately addressed these concerns.

I was initially confused the use of the word "reconstruction." For example, Figure 1 talks about "reconstruction of an SSL training image," but the generated image looks quite different from the training image. It seems you use the word differently than recent work does [1-5], where it means producing an example that is pixel-by-pixel, or feature-by-feature, similar to the input. I suggest using a different word or clarifying the issue earlier in the paper.

I think synthetic-data experiments might strengthen the paper. They might nail down specific aspects of this phenomenon. One might better probe the types of memorized information. (Of course, easier suggested than done.)

I find the related work section adequate, but I think Section 3, "Defining Deja Vu Memorization," would benefit from clearer connections between prior work and the presented definitions. In particular, the definition at line 111 has similarities to the notion of memorization used in [2], which also attempts to capture the idea of memorization of specific information about individual examples, beyond what one can infer from the data distribution. The definition for testing methodology, at line 130, is similar to the definition of memorization used in [6,7].

[1] Balle, Borja, Giovanni Cherubin, and Jamie Hayes. "Reconstructing training data with informed adversaries." 2022 IEEE Symposium on Security and Privacy (SP). IEEE, 2022.

[2] Brown, Gavin, et al. "When is memorization of irrelevant training data necessary for high-accuracy learning?." Proceedings of the 53rd annual ACM SIGACT symposium on theory of computing. 2021.

[3] Haim, Niv, et al. "Reconstructing training data from trained neural networks." Advances in Neural Information Processing Systems 35 (2022): 22911-22924.

[4] Guo, Chuan, et al. "Bounding training data reconstruction in private (deep) learning." International Conference on Machine Learning. PMLR, 2022.

[5] Hayes, Jamie, Saeed Mahloujifar, and Borja Balle. "Bounding Training Data Reconstruction in DP-SGD." arXiv preprint arXiv:2302.07225 (2023).

[6] Feldman, Vitaly. "Does learning require memorization? a short tale about a long tail." Proceedings of the 52nd Annual ACM SIGACT Symposium on Theory of Computing. 2020.

[7] Feldman, Vitaly, and Chiyuan Zhang. "What neural networks memorize and why: Discovering the long tail via influence estimation." Advances in Neural Information Processing Systems 33 (2020): 2881-2891.

**Questions:**

Section 3.1 says the datasets A, B, and B are disjoint. Appendix A.2.1 says that A and B overlap. Furthermore, as mentioned above, my understanding is that X is a face-blurred version of ImageNet, and thus X might contain (possibly blurred) examples from A and B. Can you clarify this?

Could you lay out what the "adversary," the person conducting the privacy attack, has access to?

**Limitations:**

The authors adequately address the limitations of their work.

---

> ### Author Rebuttal · Authors · 2023-08-04
>
> We sincerely appreciate the related work references!
>
> We indeed can answer your questions pertaining to the dataset splits. We will edit the paper to make these points more precise. Please do let us know if we can provide any further clarifications.
>
> **Section 3.1 says the datasets A, B, and B are disjoint. Appendix A.2.1 says that A and B overlap.**
>
> To be clear: $A$ and $B$ are always disjoint! We are very sorry about the confusion on this crucial point due to the lack of clarity in the appendix. Here is the full picture: there are 1,281,167 training images in the ImageNet data. Within these images, only 456,567 of them have bounding boxes annotations (which are needed to compute the Deja Vu score). The private set $A$ and $B$ are sampled from these 456,567 bounding boxes annotated images in such a way that set $A$ and $B$ are disjoint.
>
> If we remove the 456,567 bounding boxes annotated images from the 1,281,167 training images, we get 824,600 remaining images without annotations which never overlap with $A$ or $B$. From this set of 800K images, we took 500k images as our public set $X$. So now, we have three non overlapping sets $A$, $B$, and $X$. Then, if we remove the 500K public set images from the 824,600 images without annotations, it leaves us with 324,600 images that are neither in $A$, $B$ or $X$. For simplicity, let us call this set of remaining 324,600 images the set $C$. Then, we have split the entire ImageNet training set into four non-overlapping splits called $A$, $B$, $C$ and $X$.
>
> When running our experiment with a small number of training images, we only use the set $A$ to train SSL_A and the set $B$ to train SSL_B and then use the set $X$ as a public set for evaluation. However, to run larger scale experiments, we use as additional training data for SSL_A and SSL_B: the images sampled from the set $C$. Here, SSL_A will still be trained on set $A$ but it will be augmented with images from set $C$. The same goes for SSL_B which will still be trained on the set $B$ but augmented with images from the set C. As such, some images sampled from the set C to train SSL_A or to train SSL_B might overlap. However, this is not an issue since the evaluation is done using only images from the bounding boxes annotated set $A$ and set $B$ which are never overlapping (and to be clear -- $A$, $B$ do not overlap with $C$).
>
> We would like to thank you again for raising the lack of clarity in the appendix. We will be much more straightforward in the next revision of the paper, we hope that the introduction of the set $C$ will help to clarify the experimental setup in the appendix. Please let us know if there are any remaining concerns on this very important point.
>
> **On face-blurred ImageNet**
>
> The face-blurred ImageNet is the same thing as ImageNet which is the same as ImageNet-1K (we inadvertently use different words to refer to the same dataset). Since this is not a different dataset, it is impossible that $X$ contains images form $A$ and $B$. We used only one dataset in our work, and that is ImageNet (in which people's faces were blurred for privacy reasons). It is helpful to know this was unclear since it is indeed foundational of our assertions of memorization.
>
> **I was initially confused the use of the word "reconstruction." For example, Figure 1 talks about "reconstruction of an SSL training image," but the generated image looks quite different from the training image. It seems you use the word differently than recent work does [1-5], where it means producing an example that is pixel-by-pixel, or feature-by-feature, similar to the input. I suggest using a different word or clarifying the issue earlier in the paper.**
>
> This is helpful feedback. It is true that we do not aim to reconstruct the images pixel-by-pixel. We may be better using "partial reconstruction" in which the aim is to reconstruct memorized information (foreground) from a given crop (background). We feel that partial reconstruction of *thousands* of training images is a contribution.
>
> **Could you lay out what the "adversary," the person conducting the privacy attack, has access to?**
>
> We do twice mention that Deja Vu memorization could imply adversarial advantage. We mean that an adversary with access to partial information of a training image (e.g. crop of background or an individual’s face) could use the model to infer the remainder of the image, possibly with the help of a public dataset like disjoint set $X$. In this case, the adversary could have access to the same information that our test does, as represented by the red box of Figure 2’s Inference Pipeline. It takes in set $X$, the background crop, and the model, and returns inferences about the remainder of the training image. We will make this clear where we mention the adversarial implications of memorization.
>
> However, it is worth noting that in this paper we focus on the detection of memorization in SSL, which is distinct from a methodical study of practical adversarial risk. We do believe our discovery of memorization has implications for privacy and is fodder for future security work, but are careful to not make strong claims quantifying the exact risk imposed by this memorization. As such, we intentionally focus this paper on identifying memorization and not on defining a practical adversary.
>
> **I find the related work section adequate, but I think Section 3, "Defining Deja Vu Memorization," would benefit from clearer connections between prior work and the presented definitions.**
>
> Thank you for taking the time to share these additional references. We will work them into our related work section to more accurately position our notion of memorization.
>
> **I think synthetic-data experiments might strengthen the paper.**
>
> Thank you for your suggestion, can you explain the type of synthetic based experiment you have in mind? Any detail on the different types of memorized information you are referring to would be helpful.

---

> > ### Comment · Reviewer_H9Sq · 2023-08-12
> >
> > Thank you for your detailed comment. Based on your clarifications about the data splitting, I am raising my score to 7.
> >
> > Either "partial reconstruction" or "reconstruction" are fine: I think the paper would benefit from more quickly pointing out the difference from prior use of that word. For instance, in the caption of Figure 1, you talk about the "association of this *specific* patch of water (pink square) to this *specific* foreground object (a black swan)." But the first use of "specific" refers to a set of pixels while the second refers to a particular type of object. I wholeheartedly agree that extracting this type of association is a contribution!
> >
> > By "adversary," I meant the same thing as you mean by "test." This text description of the red box in Figure 2 is what I was looking for. I used the word to mean "whoever is extracting information" and understand that your contributions are different from attacks on real-world systems.
> >
> > Reflecting upon synthetic data, I don't have a strong suggestion. One might place patches (such as colored squares) in the background of images and try to determine the color of the square from a patch. However, even if this is successful it's not clear how much it adds to your story, which already recovers strong information other than the label.

---

### Official Review · Reviewer_9ALm · 2023-07-05

**Soundness:** 3 good
**Presentation:** 3 good
**Contribution:** 3 good
**Rating:** 6
**Confidence:** 4

**Summary:**

Self-Supervised Learning algorithms provide a label-free alternative for representation learning and is widely used in vision, and language. In this work, the authors try to understand the information content of representations learned by SSL algorithms through the lens of memorization. In particular, they define a "deja-vu" memorization notion that measures how much a given sample from the training set can be reconstructed from a pretrained model.

The authors focus on joint-embedding-based approaches, exploring the extent of such memorization during pretraining by evaluating label-inference accuracy on peripheral crops of a subset of training images ($A$). A "deja-vu" score at p% confidence measures the gap in accuracy between a target model (trained on $A$) and a reference model (trained on a different subset $B$). The gap essentially measures the extent to which exposing particular samples during training makes it easier to predict the label from partial background information.

Building on these metrics, the authors present interesting results comparing different SSL algorithms under different training configurations. Notably, long pretraining increases the risk of memorization, while the size of the dataset doesn't is as consequential (Fig 4/5). With a sequence of ablations, the authors establish empirical findings and provide design recommendations that minimize deja-vu memorization while maintaining high performance.

**Strengths:**

Some strengths of the work are outlined below:
**Originality**: The authors examine the representation quality in SSL from a privacy perspective. The work focuses on train-time characteristics and memorization for non-generative SSL algorithms, in line with similar works in language/generative models. The authors propose a novel leave-one-out mechanism for evaluating the extent of memorization (compared to correlation) of specific training-set examples across SSL algorithms.

**Significance**: Understanding the information content of representations learned by black-box SSL algorithms is important as these algorithms become widely used in safety-critical applications. In particular, the paper makes a solid case for understanding the privacy risk induced by long-pretraining routines in existing SSL algorithms.

**Quality**: The work is well-motivated and grounded in real-world challenging questions related to measuring the quality of representations learned in SSL algorithms.

**Clarity**: The paper is well written, with definitions and pedagogical examples that motivate the core insights of the work. The authors include clear, well-labeled figures and plots accompanying their observations.

**Reproducibility**: All experiments are performed on public datasets and pretrained models and sufficient details to reproduce the core results of the paper. The authors are encouraged to release the accompanying code at their earliest convenience.

**Weaknesses:**

Some weaknesses of the work are outlined below:

**Explainability**: The authors have shared some interesting empirical findings about the effectiveness of SSL algorithms in representing data and their ability to recall examples from the training set from partial side information. Although the authors demonstrate that established methods like guillotine regularization reduce the risk of memorization, the current version of the study could benefit from more discussion on the underlying mechanisms behind these mitigation techniques.


**Questions:**

1. Although the authors primarily concentrate on joint-embedding-based techniques, it is probable that reconstruction-based methods are also susceptible to this type of memorization. To enable a fair comparison across the SSL algorithm family, it would be useful to include a baseline such as MAE.nclude a baseline such as MAE.
2. The focus of the authors is on the linear probe as a potential measure of representation quality. However, they have shown that this metric is inadequate in capturing the full extent of deja-vu memorization. It would be valuable to determine if this inconsistency still exists even when the model is fine-tuned, which involves adapting the backbone as well, in an end-to-end approach.

**Limitations:**

The paper explores the privacy-risk profile of SSL algorithms through the lens of training-set memorization. The authors are strongly encouraged to discuss their proposed algorithm's limitations (including assumptions and failure modes).

---

> ### Author Rebuttal · Authors · 2023-08-04
>
> Thank you for giving our paper a detailed read, as is clear from you thorough summary! We are happy to hear the strengths you saw in our work, and try to address some of your primary questions below. Please let us know if you have any follow up questions.
>
> **Explainability: [...] Although the authors demonstrate that established methods like guillotine regularization reduce the risk of memorization, the current version of the study could benefit from more discussion on the underlying mechanisms behind these mitigation techniques.**
>
> In this work, we wanted to alert the research community about the Deja Vu phenomenon and show some situations in which it is amplified when doing SSL training. It is not clear yet what are the exact underlying mechanisms inducing this phenomena. We are keen to study this in follow up work.
>
> The guillotine regularization authors have shown that most of the over-fitting with respect to the downstream task lies in the last layers of the network. In consequence, when training an SSL model, the last layers will be heavily biased (because of the training criteria) to match the representations of different patches into the same embedding. But this is not the case for the intermediate layers of the network which are free to assign different embeddings to different crops of a given image (which might also be better for generalization).
>
> However, there are still some mysteries: we do not know why the BYOL criterion is so much more robust with respect to Dejavu memorization in comparison to Barlow Twins or VICReg. Clearly, what causes Deja Vu is subtle and complex. We believe further work would be needed to fully assess the extent of this phenomena and its origin.
>
> **It would be useful to include a baseline such as MAE**
>
> This work focuses on introducing a new modality of memorization in SSL (Deja Vu memorization) and demonstrating it empirically in SSL joint-embedding models. We considered including MAE in our experiments, but it is unclear how to make a fair comparison with those models. MAE is trained with a decoder which generates images in pixel space. The decoder is directly trained to reconstruct training examples given patches, so there is no need to introduce another reconstruction method.
>
> This being the case, measuring Deja Vu with MAE would require an entirely different methodology than our KNN-based decoder scheme. Hence for this work, we focus on joint-embedding methods. We will modify the introduction to clarify our focus on joint embedding models for the scope of this work.
>
> **It would be valuable to determine if this inconsistency still exists even when the model is fine-tuned, which involves adapting the backbone as well, in an end-to-end approach.**
>
> The extent of Deja Vu memorization in a fine-tuning setting is an interesting question. We ran experiments with fine-tuned models. In this setup, we used a pretrained VICReg and fine-tune it for 20 epochs (For both SSL_A and SSL_B). We present the results in the table below:
>
> | \ Epoch | 0 | 2 | 4 | 6 | 8 | 10 | 12 | 14 | 16 | 18 | 20 |
> |---|---|---|---|--|---|---|---|--|---|---|---|
> | Train Acc. | 60.0 | 67.2 | 73.5 | 78.0 | 81.5 | 84.9 | 87.1 | 88.9 | 90.2 | 91.5 | 91.8
> | KNN Acc. | 40 | 33 | 35.8 | 36.6 | 37.4 | 37.5 | 37.9 | 38.1 | 38.2 | 38.4 | 38.7
> | DejaVu Score | 33.2 | 23.4 | 24.7 | 25.9 | 26.5 | 27.3 | 28 | 28.4 | 31.2 | 29.9 | 32.6
>
> ---
>
> Our tests show that the training accuracy increase significantly however the KNN accuracy does not change much. It is also interesting to note that just immediately after the few first epoch of fine-tuning the deja vu score decrease significantly but it does go up again later in the training. SSL joint-embedding models are often used in a frozen setting (in contrast to MAE type of model who are often fine-tuned), however, we will gladly add these fine-tuning results in the appendix. Please let us know, if there is anything else you would want to see in this setup.

---

### Official Review · Reviewer_2FNq · 2023-07-06

**Soundness:** 3 good
**Presentation:** 3 good
**Contribution:** 2 fair
**Rating:** 6
**Confidence:** 4

**Summary:**

This work investigates to what degree self-supervised models exhibit memorization of particular samples in the training set. The authors demonstrate that a simple crop of an image, only comprised of the background, is enough for an adversary to learn the associated class label of the foreground object. Moreover, a conditional generative model can be trained, allowing the adversary to reproduce a very similar-looking image compared to the original. Such memorization capacity can lead to privacy issues and the authors demonstrate that such behaviour occurs in a broad range of models, while at the same time, this effect is not visible in common metrics such as probing accuracy.


**Strengths:**

1. The paper is very well written and builds upon the simple but very interesting idea of memorization in self-supervised models. Such an investigation seems timely, given the similar recent efforts in large language models.
2. The experiments are very well designed, clearly disentangling the notion of correlation (i.e. actually learning something generalisable about the sample) and pure memorization (i.e. only memorizing the specifics of the sample). The splitting of the training data into two sets and training two separate models is a very neat way to measure this effect, which would otherwise remain very difficult to disentangle.
3. It is very interesting that supervised and self-supervised methods behave very differently in terms of memorization. In general, studying the difference in the two learning approaches is very worthwile and timely.


**Weaknesses:**

1. I struggle to really see the privacy concern here. How likely is it really for someone to have access to only a crop of an image, while not being able to access the full image? The authors present the example of an image of a face of a person, which could then be potentially used to infer where that person is. But again, in what situation do I have access to the face of an image without the rest of it? As the authors also demonstrate, such an image has to be present in the training data, it is not enough to simply use another photo of a person and then the model reveals another image of the same person. ﻿I also believe that this scenario is different from the language model case, where simply “guessing” a good query as input for the model to complete seems to be more realistic due to the smaller search space and potential knowledge of similar queries. I hope the authors can comment on this.
2. This is more minor but while I really like the experimental setup, I’m still not sure if memorization and generalization are completely disentangled. Assume that you have a model that clearly memorizes images but also performs very poorly at the downstream task, i.e. its linear probe is around random guessing. Due to the complete lack of generalization, it will be tough to establish memorization since the nearest neighbours will all share no correlation. So from a pure theoretical perspective, this seems a bit unsatisfying. Of course in practice, no generalization is usually not the case, so from a practical viewpoint this makes complete sense.


**Questions:**

What happens if you perform the nearest neighbour class inference on the training set (excluding the image itself)? This would circumwent the issue outlined above, but of course defeats the practicality of the method. Still seems like an interesting experiment though.


**Limitations:**

The authors have addressed the limitations of their work.

---

> ### Author Rebuttal · Authors · 2023-08-04
>
> We are glad to hear that you found our work timely, and our experiments interesting. Please let us know if our responses below answer your primary questions.
>
> **I struggle to really see the privacy concern here. How likely is it really for someone to have access to only a crop of an image, while not being able to access the full image?**
>
>  Thank you for your comment.
>
> We would like to emphasize that this paper centers on finding systematic memorization of training data by SSL models. We do suggest that memorization could imply privacy risks, but a rigorous study of practical adversaries would require a complete followup work, and is not a primary focus of the work (the discussion around privacy is only a few lines in the paper). If there are any sentences in the paper that seemed to misconstrue this, please let us know and we will be glad to clarify the contributions.
>
> We acknowledge that the claims made in the paragraph on the privacy risks we presented in the related work (and also the sentence in the introduction about associating a face to an activity) might have been too strong without providing further privacy analysis. You are right that the comparison with NLP is not as straightforward since it is easier to find a sequence of words that might have been in the training set than a sequence of pixels. We are planning to update the paper accordingly to ensure that no misleading assertions or comparisons.
>
> To provide a simple example of disclosure as to why we made the analogy with NLP earlier, consider an individual’s public social media profile picture that is a crop of their face taken from a photo of them on vacation. If the full (private) image was used in SSL model training, the public profile face crop could be used to query the model and recover information about the remainder of the image, disclosing the vacation location. However, we recognize now that this potential setup might be more restricted than the ones used in NLP.
>
> We hope that modifying the misleading sentences around the privacy implications will solve your concerns.
>
> **As the authors also demonstrate, such an image has to be present in the training data, it is not enough to simply use another photo of a person and then the model reveals another image of the same person.**
>
> A core component of studying memorization is showing how models retain excessive information about training set examples. It would be interesting to see how this changes for slight variations of training set examples, and we will consider this in follow up work.
>
> **I’m still not sure if memorization and generalization are completely disentangled [...if the model’s] linear probe is around random guessing [...] it will be tough to establish memorization since the nearest neighbours will all share no correlation. What happens if you perform the nearest neighbour class inference on the training set (excluding the image itself)?**
>
> There are many ways a model could memorize its training data. We do not claim that Deja Vu completely disentangles generalization from memorization in Self-Supervised Learning.
> We claim that one modality of memorization in SSL can be revealed with our Deja Vu method.
> As you mentioned, a model could probably store a lot of information about training examples in its weights without producing meaningful embeddings. In this work, we are focusing on model’s embeddings for which we suppose they are meaningful (at least enough to give good KNN performance). Providing a more theoretical perspective on memorization might also be challenging since there might be many ways to define memorization in a DNN. In addition, defining generalization in a Self-Supervised setting is not straightforward (are you considering generalization on a pretext task or a downstream task, if so which ones ?)
>
> Concerning using the training set $A$ instead of the public set $X$, it would indeed defeat the purpose of removing training set access from the test. Moreover, using the training set for decoding would not quite disentangle memorization from performance, since it still requires the encoding of similar training set images (e.g. swans) to be close together. However, we still ran the experiment as requested and found a small drop in the DejaVu Score. We used our VICReg model which shown a DejaVu score of 33.2 and replace the public set $X$ by the set $A$ when doing KNN (from which we excluded) the image itself (for each case), and got a DejaVu score of 29.7 (the differences could be explain mostly because the set $A$ is much smaller than the set $X$). We think that adding such results in the paper might add more confusion than needed, however if you think this is an important point that should be in the paper, we would gladly further discuss it.

---

> > ### Comment · Reviewer_2FNq · 2023-08-16
> >
> > I thank the authors for engaging with my comments!
> >
> > **Privacy concern:** Makes sense to me, re-writing this part of the paper and honestly stressing the difference to the NLP scenario would clarify things a lot without diminishing the contributions in my opinion. I completely agree that the discovered phenomenon is interesting in itself.
> >
> > **Recognizing test image:** If the authors rewrite the privacy section accordingly, there is also no need for such an experiment in this work.
> >
> > **Memorization vs generalization:** Thanks for running this experiment, I see the concerns of the authors now and agree that this might only foster confusion since the setting is already a bit complex. It is still encouraging though that this experiment results in very similar memorization scores.
> > \
> > \
> > \
> > In light of the answers and conditional on the re-writing of the section regarding privacy concerns, I have raised my score.

---

### Author Rebuttal · Authors · 2023-08-04

We would like to thank all reviewers for their thoughtful remarks on our work. Reviewer H9Sq raised an important clarification, asking whether sets $A$ and $B$ are disjoint. We want to clarify for everyone that indeed our set $A$ and $B$ are always disjoint (which is fundamental in our analysis).

We also want to highlight that the core of our contribution is the detection of memorization in SSL, which is distinct from a methodical study of practical adversarial risk. We do believe our discovery of memorization might have implications for privacy and is fodder for future security work, but are careful to not make strong claims quantifying the exact risk imposed by this memorization. As such, we intentionally focus this paper on identifying memorization and not on defining a practical adversary. We recognize that the claims in our paragraph on the privacy implications of Deja Vu memorization might have been too strong without providing further empirical evidence. We will be careful to update this paragraph accordingly.

We were pleased to see the thorough and detailed summaries written by each reviewer, underscoring their careful read of the work. We have attempted to respond to each of your primary questions below. Please respond if you have any followup questions. We are eager to make our contributions as clear as possible, and your help is invaluable!

---

### Decision · Program_Chairs · 2023-09-21

**Decision:**

Accept (poster)

**Comment:**

This paper raises an interesting question studying memorization in SSL: this paper shows that it is possible to predict the foreground object of an image when given only the background, something that should be impossible if the model did not memorize. The reviewers universally liked the paper and after the back-and-forth with authors I agree the paper is high quality and should be accepted.